# Inhibition of the succinyl dehydrogenase complex in acute myeloid leukemia leads to a lactate-fuelled respiratory metabolic vulnerability

Ayşegül Erdem[1,2], Silvia Marin [2,3,4], Diego A. Pereira-Martins [1,5], Marjan Geugien[1], Alan Cunningham[1], Maurien G. Pruis[1], Isabel Weinhäuser[1,5], Albert Gerding[6,7], Barbara M. Bakker[7], Albertus T. J. Wierenga[1,6], Eduardo M. Rego[5], Gerwin Huls[1], Marta Cascante [2,3,4,8] & Jan Jacob Schuringa [1,8 ✉]

Metabolic programs can differ substantially across genetically distinct subtypes of acute myeloid leukemia (AML). These programs are not static entities but can change swiftly as a consequence of extracellular changes or in response to pathway-inhibiting drugs. Here, we uncover that AML patients with *FLT3* internal tandem duplications (*FLT3*-ITD[+]) are characterized by a high expression of succinate-CoA ligases and high activity of mitochondrial electron transport chain (ETC) complex II, thereby driving high mitochondrial respiration activity linked to the Krebs cycle. While inhibition of ETC complex II enhances apoptosis in *FLT3*-ITD[+] AML, cells also quickly adapt by importing lactate from the extracellular microenvironment. $^{13}C_3$-labelled lactate metabolic flux analyses reveal that AML cells use lactate as a fuel for mitochondrial respiration. Inhibition of lactate transport by blocking Monocarboxylic Acid Transporter 1 (MCT1) strongly enhances sensitivity to ETC complex II inhibition in vitro as well as in vivo. Our study highlights a metabolic adaptability of cancer cells that can be exploited therapeutically.

[1] Department of Experimental Hematology, University Medical Center Groningen, University of Groningen, Hanzeplein 1, 9700 RB Groningen, The Netherlands. [2] Department of Biochemistry and Molecular Biomedicine, Faculty of Biology, University of Barcelona, Avda. Diagonal 643, Barcelona 08028, Spain. [3] CIBER of Hepatic and Digestive Diseases (CIBEREHD), Institute of Health Carlos III, 28029 Madrid, Spain. [4] Institute of Biomedicine of University of Barcelona, 08028 Barcelona, Spain. [5] Hematology Division, LIM31, Faculdade de Medicina, University of São Paulo, São Paulo, SP, Brazil. [6] Department of Laboratory Medicine, University Medical Center Groningen, University of Groningen, Hanzeplein 1, 9700 RB Groningen, The Netherlands. [7] Laboratory of Pediatrics, Section Systems Medicine of Metabolism and Signaling, University Medical Center Groningen, University of Groningen, Groningen, The Netherlands. [8] These authors jointly supervised this work: Marta Cascante, Jan Jacob Schuringa. ✉email: j.j.schuringa@umcg.nl

Acute myeloid leukemia (AML) is a stem cell disease that occurs mostly in adults upon the abnormal transformation of hematopoietic stem/progenitor cells (HSPCs) in the bone marrow. Genetic aberrations lie at the heart of leukemia, and the majority have now been identified[1,2]. These and other studies uncovered the heterogeneity of the disease, whereby the combined set of mutations that give rise to AML can differ substantially between patients, and even within patients genetically distinct clones can co-exist[3–7]. As a result of the different mutational spectra, AML patients present a heterogeneous transcriptomic, epigenomic, and proteomic landscape, which leads to a variety of transcriptional networks, cell biological properties and drug sensitivities[7,8]. The complex and diverse mutational architecture of AML, which also evolves over time or as a consequence of treatment, creates one of the biggest challenges in AML therapy.

Metabolism has emerged as a targetable entity that might provide alternative means for cancer therapy. AML blast cells can be highly glycolytic, which correlates with poor prognosis, and inhibition of glycolysis enhances cytotoxicity of Cytarabine (Ara-C)[9,10]. Deletion of several glycolysis enzymes delayed leukemia progression[11]. In contrast, recent papers also indicated that in certain subtypes of AML leukemic stem cells (LSC) might in fact be particularly sensitive to OXPHOS inhibition[12–15]. Inhibition of the mitochondrial electron transport chain (ETC) complex I impaired proliferation and induced apoptosis in acute myeloid leukemia[16], in particular in those AMLs characterized by high OXPHOS dependency[16].

These studies have suggested that the heterogeneity in AML also extends to their metabolic profiles. A thorough understanding of how mutations that drive specific transcriptional programs might underlie such differences in metabolism is needed. Genetic abnormalities that involve internal tandem duplications or tyrosine kinase domain mutations in the FLT3 receptor gene (FLT3-ITD, FLT3-TKD) occur in ~30% of AML patients and are associated with poor prognosis and poor response to chemotherapy[17,18]. FLT3-ITD mutations drive distinct gene expression signatures[19], FLT3-ITD+ AMLs are characterized by increased levels of reactive oxygen species (ROS) that lead to increased DNA double-strand breaks (DSBs)[20] and inhibition of the FLT3-ITD imposes a specific dependency on glutaminolysis[21].

In this study, we evaluate metabolic signatures in a large cohort of AML patients with FLT3-ITD by combining quantitative proteome analyses with functional metabolic approaches. We identify Electron Transport Chain (ETC) complex II as a specific targetable determinant of the OXPHOS state of AML. We find that under steady-state conditions cells can already utilize lactate as carbon source to drive the TCA cycle, and blocking ETC complex II makes the cells more sensitive to inhibition to MCT1-mediated exchange of lactate with the environment. Detailed insights into metabolic programs within specific genetic subtypes of AML will not only improve AML risk-stratification and allow tailored metabolic therapy approaches but will also increase our understanding of metabolic adaptation upon therapeutic interventions.

## Results

### FLT3-ITD mutant AMLs display distinct OXPHOS-driven metabolic signatures compared to FLT3-wt AMLs. 
Previously, we identified that considerable metabolic heterogeneity exists between different subtypes of AML, whereby FLT3wt AMLs, with high expression of Pyruvate Dehydrogenase Kinase 1 (PDK1) displayed more glycolytic phenotypes and FLT3-ITD AMLs were more OXPHOS-driven[22]. In order to uncover the molecular mechanisms underlying the OXPHOS metabolism of FLT3-ITD

AMLs we first characterized FLT3-ITD-specific protein expression profiles by analyzing our label-free quantitative proteome datasets of the stem/progenitor-enriched CD34+ fraction of FLT3-wt patients ($n = 26$) and FLT3-ITD+ patients ($n = 15$) (Supplementary Fig. 1a and Supplementary Data 1 and 2)[7]. Gene ontology (GO) analyses revealed that upregulated proteins in FLT3-ITD+ AML were significantly enriched for various metabolism-related GO terms, in particular those related to the respiratory electron transport chain, the Krebs cycle (tricarboxylic acid (TCA) cycle), and lipid metabolism (Supplementary Fig. 1b). Some of the overexpressed proteins in CD34+ blasts derived from AML patients with FLT3-ITD are linked to the GO terms NADH regeneration/oxidation and respiratory electron transport chain (Supplementary Fig. 1c). These included Succinate-CoA Ligases (SUCLG1/2) and several others that are involved in NADH regeneration, regulate mitochondrial respiration-dependent energy production, and control ETC activity and the TCA cycle (Fig. 1a and Supplementary Data 2).

SUCLG and SDH (Succinate dehydrogenase, complex II of the ETC) proteins are functionally linked, whereby SUCLG proteins within the Krebs cycle generate succinate that in turn is metabolized by SDH to generate FADH 2 and consequently ubiquinone reduction. CD34+ AML blasts derived from FLT3-ITD+ patients expressed significantly higher levels of SUCLG1 and 2, although also a subset of FLT3-wt patients expressed high levels of ETC complex II proteins (Fig. 1a). We noted that the FLT3-wt subgroup with high levels of SUCLG2 was genetically distinct from the FLT3-wt group with low levels of SUCLG2. In our proteome, we observed that all NPMcyt and inv16 AML subtypes resided within the FLT3-wt/SUCLG2high group (Supplementary Fig. 1d). Pearson coefficients were calculated using the quantitative proteome comparing the FLT3-WT/SUCLG2high group with the FLT3-WT/SUCLG2low group, and the ranked list was used for GSEA which indicated significant enrichment for NPM1mut signatures, inv16 signatures, and MLL-rearranged AML signatures (Supplementary Fig. 1e). Next, we independently confirmed these data using the MILE AML transcriptome dataset (GSE13159) and healthy HSPCs (GSE42519) and observed that indeed the inv16 and MLL-rearranged AML subtypes express significantly higher levels of SUCLG2 compared to other AML subgroups (Supplementary Fig. 1f). Finally, we correlated expression of genes associated with glycolysis, TCA cycle, and glutaminolysis-associated with overall survival. A cox hazard proportional model for overall survival using continuous values for gene expression signatures retrieved from the PRECOG database[23] and a forest plot for multivariable analysis for SDHA, SDHB, SUCLG1, and SUCLG2 in shown in Fig. 1b, indicating that they act as independent predictors of overall survival in AML patients (see also Supplementary Data 1).

### The OXPHOS state correlates with high expression of SDH and SUCLG proteins and ETC complex II activity in AML. 
The mitochondrial metabolic regulatory pathways that are potentially involved in FLT3-ITD+ AMLs which we wished to functionally study further in detail are highlighted in green in Fig. 1c. Oxygen consumption rates (OCR) were functionally assessed by Seahorse in primary AML cells, indicating that OCR was significantly higher in FLT3-ITD+ AMLs compared to FLT3-wt AMLs (Fig. 1d), which correlated with SUCLG2 or SDHB expression (Fig. 1e). These data are in line with our previous observations indicating FLT3-ITD+ cell lines display higher OCR compared to FLT3-wt cell lines[22]. We transduced healthy CB CD34+ cells with lentivectors overexpressing FLT3-ITD or MLL-AF9 and observed that OCR was increased in both conditions, while ECAR levels were slightly reduced in FLT3-ITD-transduced cells and slightly

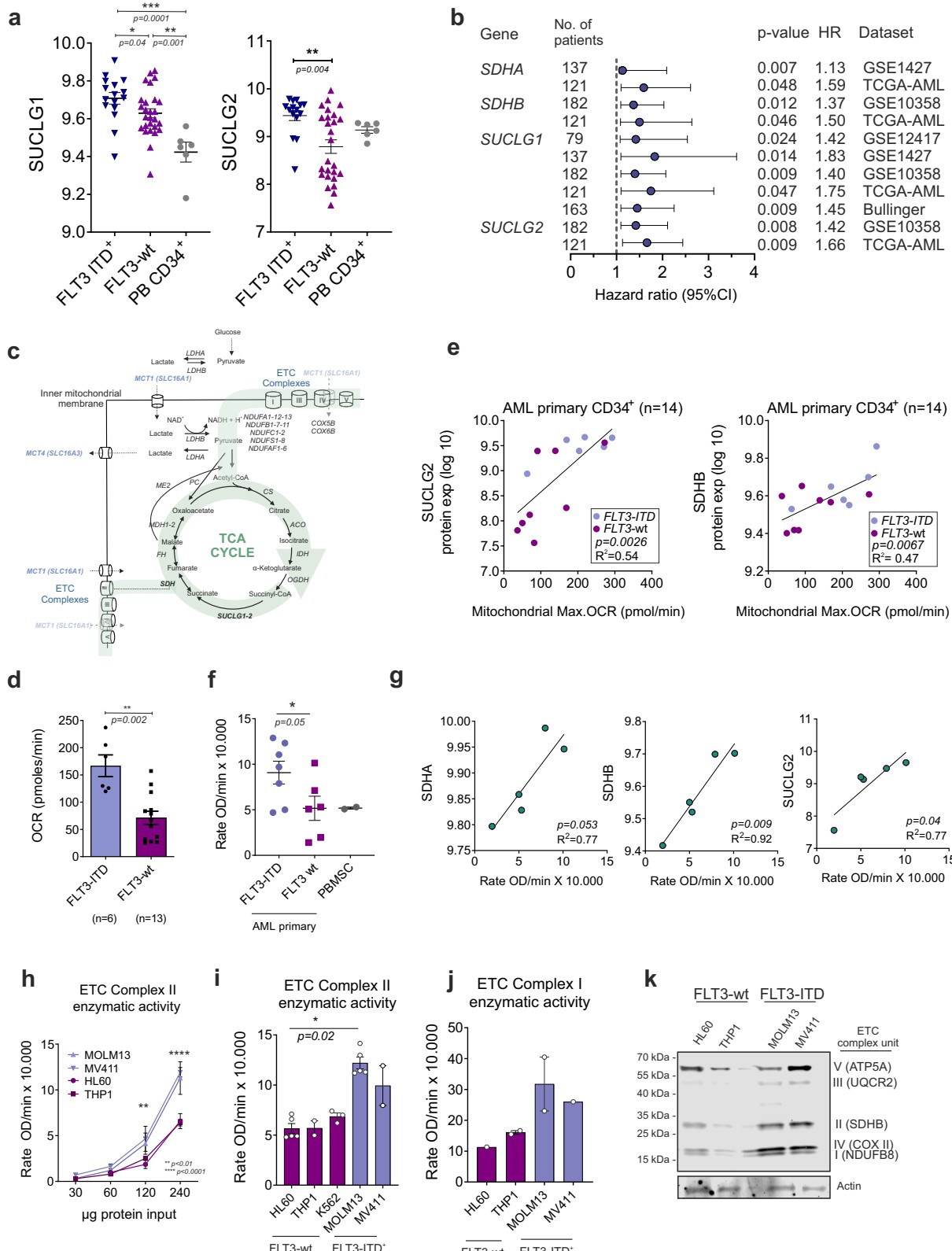

increased in MLL-AF9-transduced cells (Supplementary Fig. 2a). mRNA expression of SUCLG2 and SDHA were shown to be significantly higher in *FLT3*-ITD-transduced CB cells but not in MLL-AF9-transduced cells (Supplementary Fig. 2b). CD34[+] stem/progenitor cells from *FLT3*-ITD[+] AML patients displayed higher ETC complex II activity compared to *FLT3*-wt AMLs (Fig. 1f), which correlated with SDHA, SDHB, and SUCLG2

protein expression (Fig. 1g). The OXPHOS-driven cell lines MOLM13 and MV411 displayed higher ETC complex II activity compared to the *FLT3*-wt glycolytic line HL60 and *FLT3*-wt K562 cells (Fig. 1h, i). A similar trend was seen for ETC complex I activity, although this did not reach significance (Fig. 1j). Interestingly, the MLL-AF9-driven OXPHOS cell line THP1 displayed ETC complex I/II activity levels comparable to HL60 (Fig. 1h–i).

**Fig. 1 The OXPHOS state correlates with high expression of SDH and SUCLG proteins and ETC complex II activity in AML. a** Protein expression levels (log10) of SUCLG1 and SUCLG2 in CD34$^+$ *FLT3*-ITD$^+$ ($n = 15$), *FLT3*-wt ($n = 26$) AML patients and in healthy peripheral blood mobilized stem cells (PBSC) ($n = 6$) (mean $+/-$ SEM). **b** Forest plot for multivariable analysis for *SDHA*, *SDHB*, *SUCLG1*, and *SUCLG2* as independent predictors of overall survival in AML patients. Cox hazard proportional model for overall survival using continuous values for gene expression signatures from the PRECOG database. Hazard ratios (HR) > 1 indicates that increasing values for continuous variable or the first factor for categorical variables has the poorer outcome. HR and their respective 95% confidence interval are indicated with blue circle and a line, respectively. Public datasets were described by the title of gene series expression (GSE) or by the code of the previous studies (TCGA: $n = 121$; GSE1427: $n = 137$; GSE10358 $n = 182$; GSE12417 $n = 79$; Bullinger: $n = 163$). **c** Mitochondria-driven energy flux through TCA-cycle and ETC complexes in *FLT3*-ITD$^+$ AMLs. Functionally more active metabolic pathways are highlighted in green. **d** Basal OCR levels in CD34$^+$-sorted *FLT3*-ITD$^+$ ($n = 6$) and *FLT3*-wt ($n = 13$) AML primary samples. Each dot represents an independent patient sample, mean of technical quadruplicates. **e** Correlation of SUCLG2 and SDHB protein expression (log10) versus maximal OCR in CD34$^+$ AML primary samples. **f** ETC complex II enzymatic activity levels in CD34$^+$-sorted *FLT3*-ITD$^+$ ($n = 7$) and *FLT3*-wt ($n = 6$) AML and healthy PBMSCs ($n = 2$). Each dot represents an individual sample (mean of technical duplicates) (mean $+/-$ SEM). **g** Correlation of enzymatic ETC complex II activity compared to SDHA, SDHB, and SUCLG2 protein expression (log10) in CD34$^+$ AML primary samples ($n = 5$). **h** ETC complex II activity in different protein amounts and (**i**) in 240 µg protein input of AML cells. Mean $+/-$ SEM of independent replicates, in technical duplicates. **j** ETC complex I activity in (240 µg protein) AML cells ($n = 1$ or 2). **k** Western blot analysis of ETC subunits; complex I (NDUFB8,18 kD), II (SDHB, 29 kD), III (UQCRC2,48 kD), IV (COX II, 22 kD), and V (ATP5A, 54 kD) in AML cells. An equal amount of protein loaded per cell line and Actin is the loading control and the image is a representative of independent triplicates. **e**, **g** linear regression analysis, (**a**, **d**) Student's *t* test (two-sided) or (**f**, **i**) one-way, (**h**) two-way ANOVA for multiple comparisons.

At the protein expression level, higher levels of ETC complexes I, II, and IV were detected in *FLT3*-ITD$^+$ AML cell lines compared to *FLT3*-wt cell lines (Fig. 1k). These data nicely align with our previous observations where in an individual patient *FLT3*-ITD$^+$ and *FLT3*-wt subclones could prospectively be identified and isolated[7], whereby The *FLT3*-ITD$^+$ subclone also expressed higher levels of ETC complexes I, II, and IV[22]. Lentiviral over-expression of the *FLT3*-ITD in the *FLT3*-wt AML cell lines HL60 or TF1 was sufficient to increase ETC complex II activity (Supplementary Fig. 2c). Collectively, these data indicate that a higher OXPHOS state correlates with high expression of SDH and SUCLG proteins, which coincides with the higher OCR and ETC complex II activity seen in *FLT3*-ITD$^+$ AMLs.

**Inhibition of the mitochondrial succinyl dehydrogenase complex is a vulnerability in *FLT3*-ITD$^+$ AML.** With the aim to further improve personalized treatment strategies in AML based on distinct OXPHOS signatures, we evaluated the efficacy of ETC I (rotenone) and several ETC II inhibitors (thenoyltri-fluoroacetone (TTFA) and 3-Nitropropionic acid (3-NPA)) in *FLT3*-ITD$^+$ versus *FLT3*-wt AMLs and healthy CB-derived CD34$^+$ cells. Within 24 h a strong increase in apoptosis was seen in the *FLT3*-ITD$^+$ AML cell lines MV4–11 and MOLM13 upon incubation with ETC complex I/II inhibitors, while moderate effects were seen in *FLT3*-wt THP1 and HL60 cells (Fig. 2a). CB-derived CD34$^+$ cells were also not sensitive to ETC complex II inhibition although a slight increase in apoptosis was seen upon ETC complex I inhibition (Fig. 2b). In primary CD34$^+$ AML cells, the response to ETC complex II inhibition was heterogeneous (Fig. 2c), whereby *FLT3*-ITD$^+$ AMLs were within the most sensitive group (Fig. 2d). These data are in line with previous observations indicating that *FLT3*-wt/PDK1$^{high}$ primary AML samples are not very sensitive to complex II inhibition[22]. Twenty-four hours of treatment with the PDK1 inhibitor DAP, to drive cells from glycolysis into an OXPHOS state, did not significantly reduce the viability of *FLT3*-ITD$^+$ AMLs as compared to ETC complex II inhibition (Supplementary Fig. 2d), again in agreement with previously published data[22]. Cell viability across a panel of independent non-malignant CD34$^+$ cells from normal bone marrow (NBM) ($n = 1$), PBSC ($n = 5$), and CB ($n = 5$) was not affected by TTFA (Fig. 2e). To functionally show that TTFA inhibits ETC complex II activity we treated MOLM13, HL60, and HL60 cells transduced with *FLT3*-ITDs. The enhanced activity of ETC complex II in *FLT3*-ITD$^+$ MOLM13 cells and in HL60 cells overexpressing *FLT3*-ITDs could be reduced by TTFA (Fig. 2f

and Supplementary Fig. 2e). We also tested the potential apoptotic effects of ETC complex II inhibitors combined with the *FLT3*-ITD inhibitor AC-220 (quizartinib) in MOLM13 cells and showed that apoptosis was significantly further enhanced after co-inhibition (Supplementary Fig. 2f).

As an independent validation to evaluate SUCLG activity in AML subtypes, we used SUCLG1 and SUCLG2-directed shRNAs to downregulate their expression in *FLT3*-wt (HL60 and K562) and *FLT3*-ITD$^+$ AMLs (MOLM13 and MV411) (Supplementary Fig. 2g). Downregulation of SUCLG1 or SUCLG2 significantly reduced the growth of *FLT3*-ITD$^+$ MOLM13 and MV411 cells (Fig. 2g and Supplementary Fig. 2h, respectively), while the proliferation of *FLT3*-wt HL60 and K562 cells were unaffected (Fig. 2h and Supplementary Fig. 2i, respectively). TTFA-induced apoptosis levels were significantly lower after knockdown of SUCLG1/2 s in *FLT3*-ITD$^+$ AML cells (Fig. 2i, j).

**ETC complex II inhibition results in decreased mitochondrial respiration which can be rescued by the addition of exogenous lactate.** To investigate the metabolic consequences following 24 h of ETC complex II inhibition (Fig. 3a), we first measured mitochondrial respiration and glycolytic activity by Seahorse. The OCR was markedly reduced in *FLT3*-ITD$^+$ AML cell lines MOLM13 and MV4–11 upon 24 h TTFA treatment, but not significantly in *FLT3*-wt cells (Fig. 3b). At earlier time points, 6 h TTFA treatment with 100 µM was already sufficient to reduce OCR levels while apoptosis was still not observed at that concentration (Supplementary Fig. 2j). ECAR was also reduced upon TTFA treatment (Fig. 3c).

Downregulation of SUCLG1/2 in *FLT3*-ITD$^+$ AML cell lines MOLM13 and MV4–11 also resulted in significantly lower OCR levels (Fig. 3d), but not in shSUCLG1/2 transduced *FLT3*-wt K562 cells (Supplementary Fig. 2k). Remarkably, the extracellular acidification rate (ECAR) was also reduced after SUCLG1/2 knockdown (Fig. 3e). ECAR activity levels remained unaffected after SUCLG1/2 knockdown in *FLT3*-wt K562 cells (Supplementary Fig. 2k).

Next, we measured glucose, glutamine, and lactate concentrations in the culture media of *FLT3*-ITD$^+$ MOLM13 AML cells at 10, 24, and 32 h after ETC complex II inhibition. These studies revealed that, whereas glutamine consumption was only slightly reduced after 32 h of ETC complex II inhibition, glucose consumption was almost entirely abolished, corresponding to a strong reduction of lactate excretion (Fig. 3f). This was not simply due to a reduction in cell viability or proliferation because the

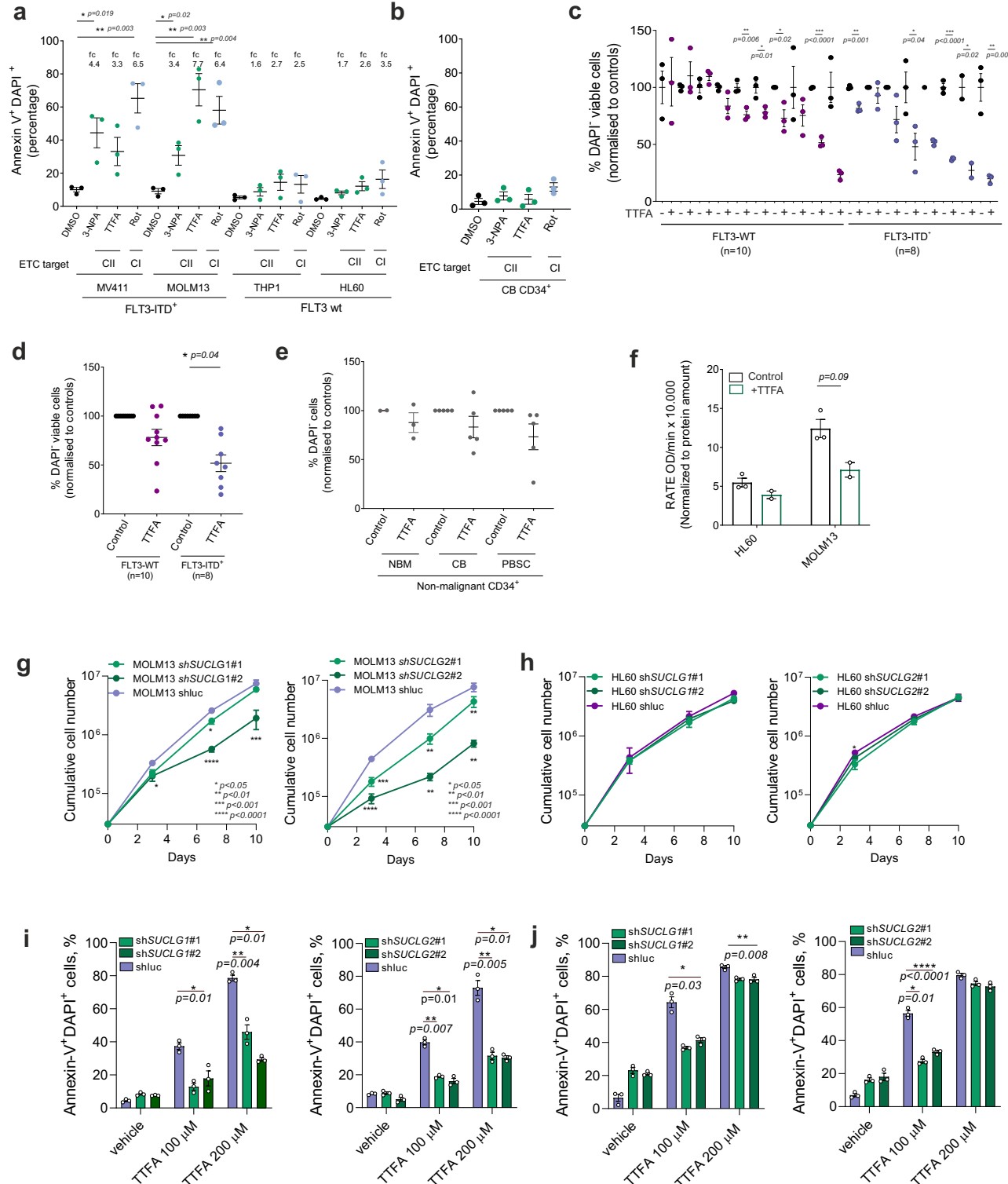

specific flux of lactate secretion, calculated considering cell number and time, was comparably reduced (Fig. 3g). These reductions in extracellular lactate were not seen when ETC complex II activity was inhibited in HL60 cells (Fig. 3g).

Lactate was shown to fuel oxidative cancer cells instead of being exported as a waste product[24]. We then wondered if the *FLT3*-ITD[+] cells might also consume lactate. Possibly, cells might rewire their metabolism and consume (more) lactate as an alternative carbon source in the non-truncated TCA-cycle branches in response to ETC complex II inhibition. We first

questioned whether the exogenous addition of lactate would rescue the apoptotic phenotype induced upon ETC complex II inhibition. Indeed, a significant reduction in TTFA-induced apoptosis was observed when either MOLM13 or MV4–11 cells were supplied with 2 mM additional lactate (Fig. 3h). This was not observed in THP1 (MLL-AF9), HL60 or healthy CB CD34[+] cells that were also intrinsically less sensitive to TTFA treatment (Fig. 3h). TTFA-induced apoptosis was also reduced in primary CD34[+] *FLT3*-ITD[+] AML cells in the presence of 2 mM exogenous lactate (Fig. 3i). We also tested whether pyruvate

**Fig. 2 Inhibition of the mitochondrial succinyl dehydrogenase complex is a vulnerability in *FLT3*-ITD[+] AML. a** (%) Annexin V[+] DAPI[+] AML cells ($n = 2$ *FLT3*-ITD[+], $n = 2$ *FLT3*-wt) upon ETC complex II inhibitors (thenoyltrifluoroacetone 200 μM, as TTFA, and 3-nitropropionic acid 2 mM, as 3-NPA) and complex I inhibitor rotenone treatment (20 μM) for 24 h. Data presented as +/− SEM of independent measurements (each dot) measured in three technical replicates. FC = fold change in apoptosis compared to controls. For gating strategy, see Source Data. **b** (%) Annexin V[+] DAPI[+] healthy CD34[+] CB cells upon ETC complex II inhibitors (TTFA 200 μM, 3-NPA 2 mM) and complex I inhibitor rotenone treatment (20 μM) (24 h). Data presented as mean +/− SEM of three independent measurements (each dot) measured in technical triplicates. **c** (%) DAPI[−] viable CD34[+] AML primary patient cells following 100 μM TTFA (48 h) on MS5 stromal co-culture. Each dot represents mean +/− SEM of technical replicates measured from each independent primary patient sample. **d** (%) DAPI[−] viable CD34[+] AML primary patient cells ($n = 10$ *FLT3*-ITD[+], $n = 8$ *FLT3*-wt AMLs) following 100 μM TTFA treatment (48 h) on MS5 stromal co-culture. Each dot represents an independent patient sample, measured in technical triplicates (mean +/− SEM). **e** (%) DAPI[−] viable CD34[+] sorted healthy NBM ($n = 1$), CB ($n = 5$), and PBSC ($n = 5$) cells after 100 μM TTFA treatment (48 h). Each dot represents technical replicates for NBM, and mean of independent primary samples for CB and PBSC. Data was normalized to the control in each group (mean +/− SEM). **f** Colorimetric ETC complex II activity measurement in AML cells ($n = 2$) in the presence and absence of 200 μM TTFA for 24 h. Data were normalized to protein amounts. Each bar represents mean +/− SEM of two biological replicates measured in two technical replicates. **g, h** Cumulative cell growth of MOLM13 (**g**) and HL60 (**h**) cells transduced with shluc, sh*SUCLG1*#1, sh*SUCLG1*#2, sh*SUCLG2*#1, sh*SUCLG2*#2. **i, j** (%) Annexin V[+] DAPI[+] shluc, sh*SUCLG1*#1, sh*SUCLG1*#2, sh*SUCLG2*#1, sh*SUCLG2*#2 MOLM13 (**i**) and MV411 (**j**) cells in the presence and absence of TTFA treatment. **a, c, d, f** Student's *t* test (two-sided) or (**g, h, i, j**) two-way ANOVA for multiple comparisons.

could act as an alternative energy substrate to drive TCA-cycle activity, and similar to lactate, the viability was increased upon the addition of exogenous pyruvate to MOLM13 or MV4–11 cells, but not HL60 cells (Fig. 3j). Lactate can be imported via the monocarboxylate transporter 1 (MCT1, encoded by *SLC16A1*) which is expressed both at the plasma membrane as well as the mitochondrial membrane where it primarily facilitates lactate influx fueling the TCA cycle[25]. We evaluated two independent MCT1 inhibitors (α-cyano-4-hydroxycinnamate (CHC) and AZD3965), and exposure of MOLM13 cells to these inhibitors resulted in elevated levels of extracellular lactate, indeed indicating that MOLM13 cells can actively consume lactate (Fig. 3k). MCT1, COX (cytochrome oxidase, ETC complex IV), and LDH (lactate dehydrogenase) co-localize in the mitochondrial membrane to form a mitochondrial lactate oxidation complex[26]. The lactate dehydrogenase B (LDHB) isoform reduces lactate to pyruvate in cells that utilize lactate as a fuel for oxidative metabolism[27]. We observed in primary *FLT3*-ITD[+] blasts from AML patients that protein levels of MCT1 and LDHB were significantly positively correlated, opposite to what was seen in *FLT3*-wt AMLs (Supplementary Fig. 2l).

We then wished to address this further in real-time oxygen consumption measurements using Seahorse and Oroboros. As expected, inhibition of ETC complex II activity resulted in a reduction in OCR in *FLT3*-ITD[+] MV4–11 and MOLM13 cells (Fig. 4a) while this was not observed in *FLT3*-wt HL60 cells (Supplementary Fig. 3a) when TTFA was injected alone. After the addition of increasing concentrations of exogenous lactate, OCR could be restored, but not when MCT1 transporters were inhibited (Fig. 4a). This effect on OCR was noted to a lesser extent in glycolytic HL60 cells (Supplementary Fig. 3a). Similar observations were done in a panel of two primary *FLT3*-ITD[+] AML samples (Fig. 4b). These Seahorse data in which lactate was transported through the intact cell plasma membrane were then independently confirmed in Oroboros measurements where we initially permeabilized the plasma membrane to demonstrate lactate transport over the mitochondrial membrane in MOLM13 cells (Fig. 4c). First, ADP, malate, and lactate were injected to study the impact on mitochondrial oxygen consumption, after which TTFA and the MCT1 inhibitor were consecutively added to evaluate their effect on mitochondrial respiration. Finally, the ETC uncoupler FCCP was added to induce maximal respiration. Injection of TTFA reduced respiration in the absence of lactate, and respiration was also reduced when MCT1 was inhibited alone in the presence of lactate (Fig. 4c, see Supplementary Fig. 3b for real-time data). The TTFA-induced reduction in respiration was less pronounced when lactate was also present, which was not the

case when lactate influx was blocked by MCT1 inhibitors CHC or AZD3965, resulting in a further significant reduction in mitochondrial oxygen consumption (Fig. 4c). The addition of lactate or the presence of CHC or AZD3965 did not show a significant impact on mitochondrial oxygen consumption of HL60 cells when TTFA was present (Supplementary Fig. 3c, d for representative real-time data). We also compared the impact of ADP, malate, and lactate on the mitochondrial oxygen consumption rate in permeabilized CD34[+]-sorted AML patient samples and showed that *FLT3*-ITD[+] cells have higher basal and ADP-stimulated mitochondrial respiration compared to *FLT3*-wt (Supplementary Fig. 4e). *FLT3*-wt cells demonstrated lower respiration which was less impacted on by TTFA in comparison to its effects on ITD[+] cells (Supplementary Fig. 3e). These data collectively show that AML cells have the capacity to restore mitochondrial respiration when lactate is shuttled through MCT1 transporters upon inhibition of the succinyl dehydrogenase complex.

To formally prove that lactate can be utilized as a carbon source and to trace in detail how it is metabolized by leukemic cells we performed [13]C$_3$-labeled lactate-tracing studies. MOLM13 cells were grown in the presence of 8 mM or 20 mM uniformly stable isotope-labeled lactate (U-[13]C$_3$ lactate) with or without TTFA for 6 h after which metabolites were extracted for gas chromatography-mass spectrometry (GC-MS) analyses (Supplementary Data 3 for metabolite list). We first analyzed total intracellular metabolite concentrations in control cells grown without labeled lactate and compared that to cells grown with U-[13]C$_3$ lactate (8 mM), with TTFA alone, with TTFA + U-[13]C$_3$ lactate (8 mM), or with TTFA + U-[13]C$_3$ high lactate (20 mM). While the addition of labeled lactate did not result in increased intracellular lactate in the absence of the ETC complex II inhibitor, addition of TTFA resulted in a significant decrease in intracellular lactate when exogenous labeled lactate was not added (Fig. 4d). Furthermore, we observed that the succinate pool was significantly larger after TTFA treatment, validating the inhibiting effects of TTFA on ETC complex II activity, resulting in intracellular succinate accumulation (Fig. 4d). While main TCA-cycle metabolites citrate, glutamate, α-ketoglutarate (Fig. 4d), and others (Supplementary Fig. 3f) were not impacted by TTFA, we also noted significant increases in aspartate, fumarate, and malate upon ETC complex II inhibition (Fig. 4d). Intriguingly, however, while upon further addition of exogenous labeled lactate the effect of TTFA on succinate accumulation was further pronounced, the opposite was observed for aspartate and fumarate, suggesting that lactate processing is able to overcome the increase in fumarate and aspartate induced by ETC complex II inhibition (Fig. 4d).

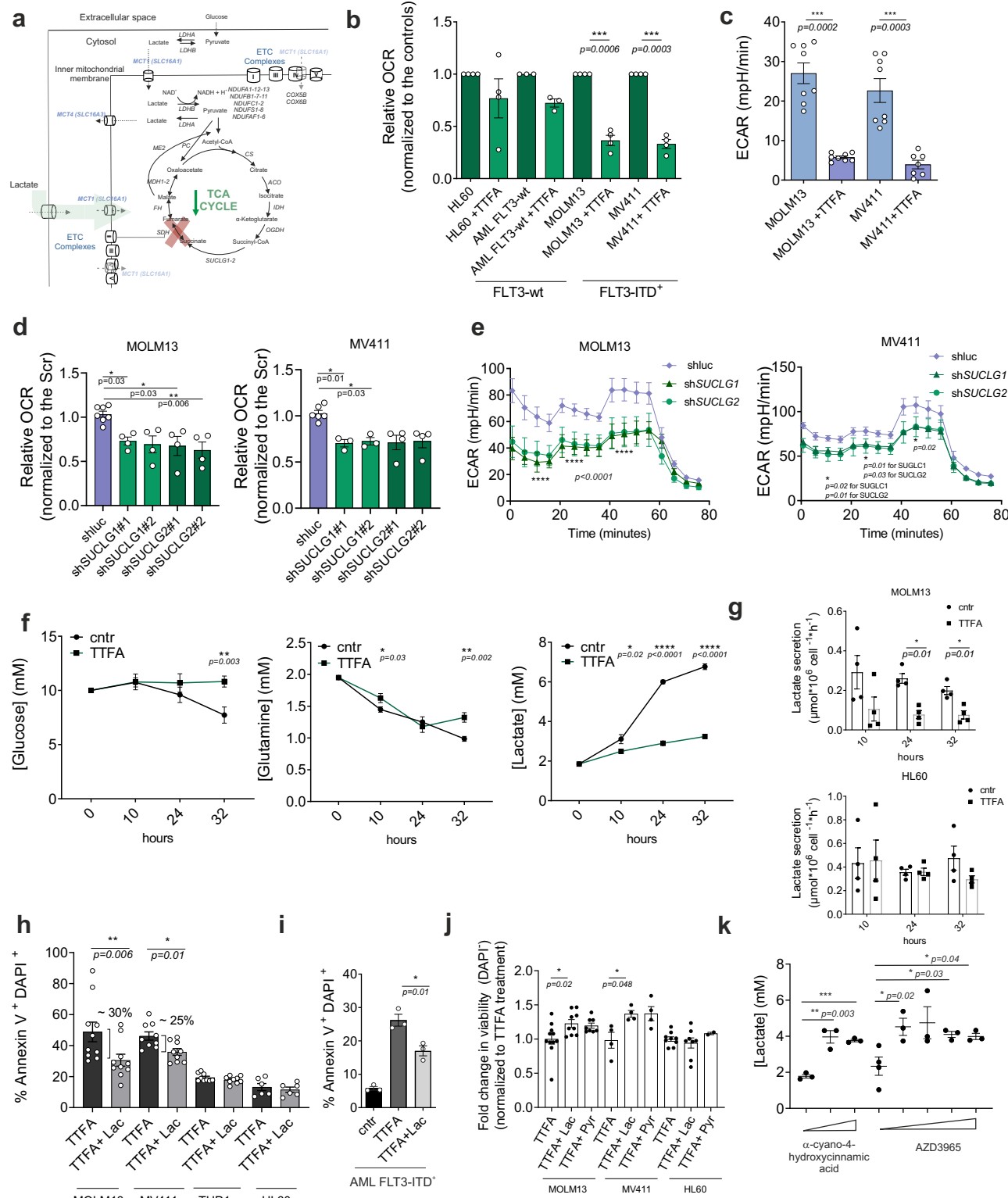

Next, we investigated label incorporation in various TCA-cycle intermediates and confirmed that lactate can indeed act as a carbon source, with the highest label incorporation in citrate and its downstream metabolites under both untreated and treated conditions (Fig. 4e, f). Upon inhibition of ETC complex II activity, a significant accumulation of the $^{13}C$ label was observed in succinate, in line with the threefold increase in succinate levels observed in TTFA treated cells and incubated with $^{13}C_3$ lactate (Fig. 4e, f). Total pools of labeled carbon incorporation from $^{13}C_3$

lactate in TCA-cycle intermediates were significantly enhanced when a higher concentration (20 mM) of exogenous lactate was added. Our mass isotopomer distribution analysis revealed that lactate was converted to pyruvate, after which it was further being metabolized through PDH (Pyruvate dehydrogenase, label incorporation indicated in purple in Supplementary Fig. 4) and PC (Pyruvate carboxylase, label incorporation indicated in green in Supplementary Fig. 4) to generate citrate that can be exported out of the mitochondria resulting in m + 3 and m + 5 labeling in

**Fig. 3 Inhibition of ETC complex II results in lower respiration and decreased lactate efflux in *FLT3*-ITD⁺ AML cells. a** Schematic representation of the metabolic outcome of mitochondrial energy flux in *FLT3*-ITD⁺ AMLs after TTFA. Functionally more active metabolic pathways are highlighted in green (mean +/− SEM). **b** Effect of TTFA (100 µM, 24 h) on oxygen consumption rates (OCR) in AML cell lines and an AML primary patient. Data normalized to controls (mean +/− SEM of three biological replicates for cell lines and $n = 1$ AML primary sample (technical triplicates are shown). **c** Effect of TTFA (100 µM, 24 h) on extracellular consumption rate (ECAR) in AML cell lines (mean +/− SEM). **d, e** OCR levels (**d**) and ECAR levels (**e**) in shluc, sh*SUCLG1*#1, sh*SUCLG1*#2, sh*SUCLG2*#1, sh*SUCLG2*#2 MOLM13 and MV411 cells. **f** Spectrophotometric real-time measurement of lactate, glutamine, and glucose concentration in the culture medium of both treated/untreated (TTFA 100 µM, 32 h) MOLM13 cells. Data represent mean +/− SEM of three biological replicates, in four technical replicates. **g** Flux of lactate secreted to the media, corrected by cell and incubation time, measured in MOLM13 and HL60 cells upon ETC complex II inhibitor (TTFA 100 µM) at 10, 24, and 32 h (mean +/− SEM, four biological replicates (each dot) measured in four technical replicates. **h** (%) Annexin V⁺DAPI⁺ AML cells ($n = 2$ *FLT3*-ITD⁺, $n = 2$ *FLT3*-wt) upon TTFA (200 µM) with/without 2 mM lactate addition in the culture medium (24 h). Each bar represents mean +/− SEM of four biological replicates (measured in technical triplicates). Percentage apoptotic cell population difference is indicated above the bars. **i** (%) Annexin V⁺DAPI⁺ in AML *FLT3*-ITD⁺ CD34⁺ cells upon TTFA (200 µM) with/without 2 mM lactate addition in the culture medium for 24 h. Each bar represents mean +/− SEM of technical triplicates. **j** Fold change in DAPI- viable AML cell lines upon TTFA (200 µM) with/without 2 mM lactate or 2 mM Pyruvate addition in the culture medium (24 h) (mean +/− SEM, each dot shows independent biological replicates. **k** Extracellular lactate concentration with/without increasing concentrations of MCT1 inhibitor CHC (α-cyano-4-hydroxycinnamic acid 400, 800 µM) and AZD3965 (25, 50, 100, 200 nM) for 24 h in MOLM13 cells. Cell viability was comparable between groups. **b, e, f, g, i, h, j, k** Student's *t* test (two-sided, Mann–Whitney), (**c**) one-way ANOVA Kruskal–Wallis test or (**d**) two-way ANOVA for multiple comparisons.

the cytosol (Supplementary Fig. 4a). Inside the mitochondria, citrate can be metabolized via the oxidative metabolism pathway through mitochondrial IDH (Isocitrate dehydrogenase) to generate α-ketoglutarate (α-KG) and consecutively succinate (resulting in m + 2 and m + 4 labeling, Supplementary Fig. 4a). However, this α-KG can also be metabolized via the reductive metabolism pathway (Supplementary Fig. 4b), resulting in label incorporation at positions m + 2 and m + 4 of citrate. This m + 2/m + 4-labeled citrate can then be converted into oxaloacetate (m + 2) and in malate, aspartate, and fumarate through counter-clockwise reductive carboxylation reactions (Supplementary Fig. 4b). This counter-clockwise reaction will generate different labeling patterns in aspartate, malate, and fumarate depending on the m + 2 (metabolized) or m + 3 (not metabolized) labeled cytosolic citrate (Supplementary Fig. 4). In this regard, while total carbon (e.g., m + 1, m + 2, m + 3, m + 4, and m + 5) labeling of metabolites from U-¹³C₃ lactate was comparable between control and treated cells (Supplementary Data 3), the total pools of m + 2 isotopologues of both malate and aspartate were significantly enriched (at ~18% and 22%, respectively) after inhibition of ETC complex II activity (Fig. 4g), indicating that this mechanism is favored in ETC complex II-inhibited cells. Although further studies are required to obtain a comprehensive detailed insight, this suggests that upon ETC complex II inhibition lactate is channeled into the TCA cycle predominantly via the cytosolic reductive carboxylation route rather than directly via the Krebs cycle (Supplementary Fig. 4b). Cytosolic reductive carboxylation rerouting was previously identified from labeled glutamine and glucose[28,29] when SDH is truncated, and was also predicted by computer modeling-based simulations in SDH deficiency[30]. Taken together, we show that AML cells can utilize lactate as a carbon source at the steady-state (Supplementary Fig. 4) resulting in extensive incorporation of uniformly labeled ¹³C₃-lactate in 40–60% of all main TCA-cycle intermediates, even when sufficient amounts of glucose and glutamine were present in the medium. When ETC complex II is inhibited, an increased exogenous pool of lactate is able to rescue respiration, and *FLT3*-ITD⁺ AML cells continue to actively metabolize and incorporate lactate, now more towards the reductive cytosolic branch of the truncated TCA cycle (Supplementary Fig. 4b).

**Dual inhibition of MCT1 transporters and ETC complex II improved treatment efficacy in vitro and in vivo.** Since inhibition of ETC complex II activity in *FLT3*-ITD⁺ AML resulted in an increasing shift towards lactate-driven OCR we wished to

exploit this therapeutically by inhibiting both ETC complex II (using 3-NPA or TTFA) and MCT1 transporters to prevent lactate import (Fig. 5a). Inhibition of MCT1 (using CHC) together with ETC complex II resulted in a strong induction of apoptosis in *FLT3*-ITD⁺ cell lines, which was much less pronounced in *FLT3*-wt cell lines (Fig. 5b). We independently confirmed the therapeutic efficacy by combining ETC complex II inhibition with the second MCT1 inhibitor AZD3965 in MOLM13 cells and observed higher apoptosis upon co-inhibition (Fig. 5c). Also, primary AML samples were evaluated when grown on bone marrow stromal cells and the strongest decreases in cell viability upon co-inhibition of SDH and MCT1 were seen in primary *FLT3*-ITD⁺ AMLs, whereby healthy control CB-derived CD34⁺ cells were not sensitive to this combination of inhibitors (Fig. 5d). In primary *FLT3*-ITD⁺ AMLs, the extracellular lactate levels decreased upon TTFA treatment, which was not observed in *FLT3*-wt AMLs or in healthy CB and PB samples (Fig. 5e and Supplementary Fig. 5a). Finally, we also wished to confirm the efficacy of ETC complex II inhibition in glycolytic (Fig. 5f–i) and OXPHOS-dependent (Fig. 5j–n) leukemias in vivo and we also tested the combination of ETC complex II and lactate transporter inhibition in vivo in *FLT3*-ITD⁺ AML (Fig. 5j). We transplanted luciferase-transduced *FLT3*-ITD⁺ MV4–11 cells that are OXPHOS-driven (Fig. 1i, k) and after confirmation of engraftment (Supplementary Fig. 5b) mice were randomly assigned to four treatment groups: vehicle, 3-NPA, CHC or combination of both. Inhibition of ETC complex II did not impact on the leukemia development of glycolytic *FLT3*-wt HL60 cells (Fig. 5g–i). Although BM AML chimerism after 3-NPA treatment was comparable to controls in both HL60 and MV4–11 transplanted mice, in contrast, 3-NPA treatment decreased disease burden in peripheral blood and resulted in a reduced spleen size in OXPHOS-dependent *FLT3*-ITD⁺ MV4–11 cells (Fig. 5k–m).

We evaluated human mtDNA (mitochondrial DNA) content from the peripheral blood after confirmation of the engraftment by luciferase intensity measurement in both HL60 and MV411-transplanted mice. Indeed, peripheral blood human mtDNA levels were shown to significantly positively correlate with luciferase intensity in mice (Supplementary Fig. 5c). At the time of comparable engraftment, *FLT3*-ITD⁺ MV411-transplanted mice showed significantly higher mtDNA levels in peripheral blood compared to HL60-transplanted mice (Supplementary Fig. 5d). Co-treatment of 3-NPA and CHC significantly reduced bone marrow human mtDNA content at time of sacrifice validating reduced tumor burden in MV411-transplanted mice

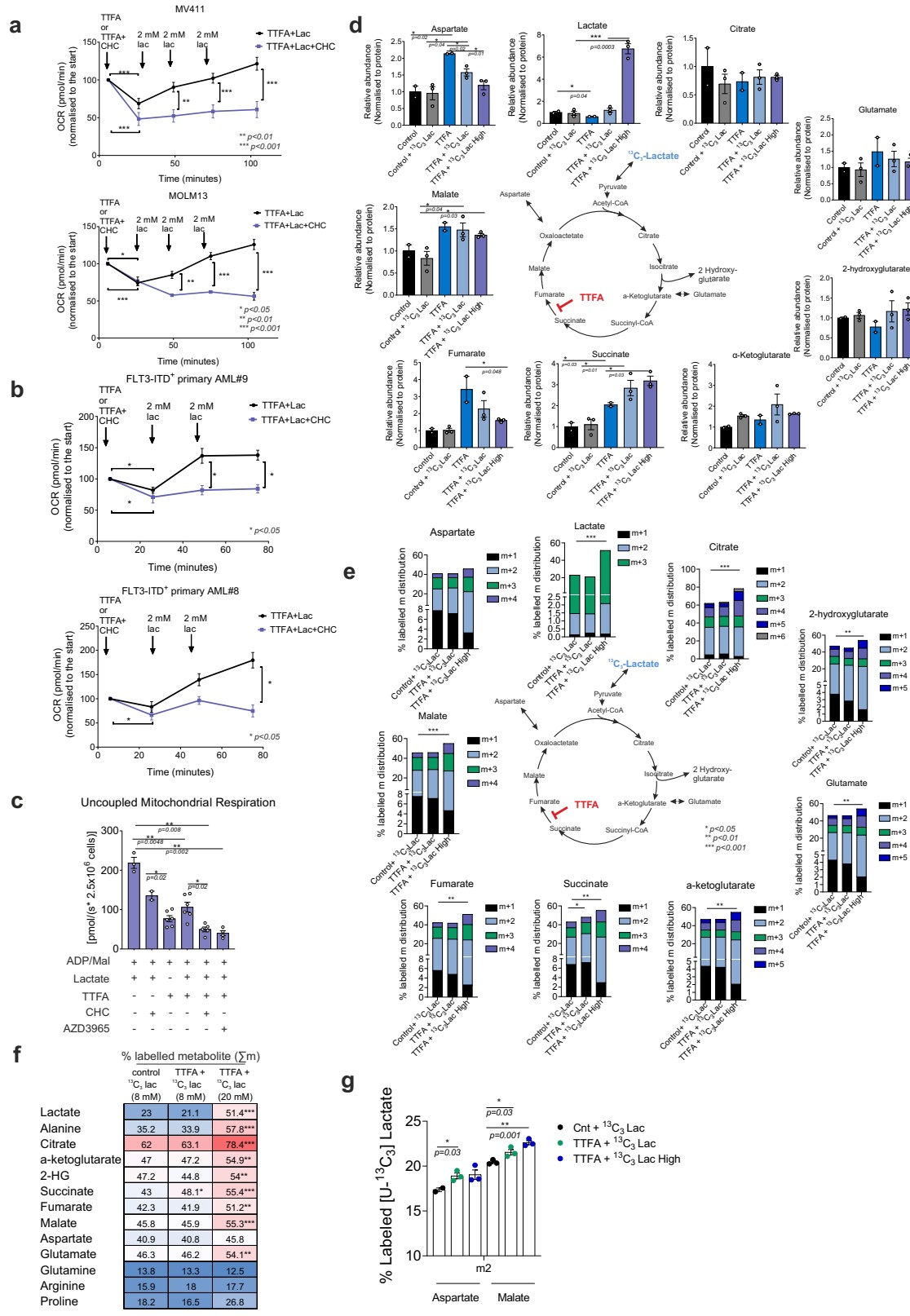

(Supplementary Fig. 5e). Mean luciferase intensity was also less pronounced at day 10 after combination treatment compared to the vehicle group (Supplementary Fig. 5f). The combination of 3-NPA and CHC treatment significantly reduced AML chimerism in PB (Fig. 5k), in the spleen (Fig. 5l) and reduced spleen weight (Fig. 5m), as well as blood leukocyte counts

(Supplementary Fig. 5g) and leukemic cell counts per femur (Supplementary Fig. 5h). While monotherapies already slightly delayed leukemia development in mice transplanted with FLT3-ITD[+] MV4–11, co-inhibition of lactate import and ETC complex II significantly further delayed leukemia onset compared to the vehicle group (Fig. 5n).

**Fig. 4 Lactate fluxes into TCA cycle and restores respiration activity of *FLT3*-ITD+ AMLs. a** Relative real-time OCR in MV411 and MOLM13 cells. Either 600 μM TTFA or 600 μM TTFA with 600 μM CHC were injected at the start of the measurement and then 2 mM sodium lactate was injected every 25 min serially. Data normalized to the initial OCR at 100. Data show mean of four biological replicates, in quadruplicate repeats (mean +/− SEM). **b** Similar to Fig. 4a, real-time OCR in two *FLT3*-ITD+ AML primary samples after CD34+ sorting followed by 24 h MS5 culture. Data show two independent biological replicates measured in quadruplicates (mean +/− SEM). **c** Real-time high-resolution respirometry measurement in MOLM13 cells in the presence/absence of sequentially injected digitonin (2.5 μg/ml), malate (2 mM) with/without lactate (8 mM), ADP (1 mM), TTFA (200 μM), CHC (400 μM), (or AZD3965 (100 nM)), FCCP (1.5 μM) in MOLM13 (2.5 × 10^6 cells/ml). After digitonin, Malate and ADP are used at the beginning and FFCP is used at the end of all measurements as major respiration substrates. Additional injection conditions are indicated below the bar graphs. Injections followed every 3–5 min after respiration was stable. Bar graphs demonstrate end-point respiration levels after FCCP was injected according to each condition combination indicated below the x axis (n = 3 biological replicates, mean +/− SEM). For real-time representative data, see Supplementary Fig. 3b. **d** Internal relative abundance of TCA-cycle associated metabolites in MOLM13. Cells were grown with/without 200 μM TTFb and with/without 8 mM and 20 mM (high) labeled [U-13C3] lactate (6 h). Ion counts for each metabolite were normalized to internal recovery standards, and to the protein amount of the cell pellets (mean +/− SEM of independent triplicates). **e** (%) Isotopologue distributions (13C enrichment in each number of carbons) of TCA-cycle associated metabolites in MOLM13. Cells were grown with/without 200 μM TTFA and with 8 mM and 20 mM (high) labeled with [U-13C3] lactate for 6 h. **f** (%) total labeled molecules (Σm) obtained from the incubation with [U-13C3] lactate in the metabolites shown in (**e**). **g** m + 2 pools of malate and aspartate from [U-13C3] lactate in MOLM13 cells. Data are mean−/+SEM, n = 3 replicates. **a–g** Student's t test (two-sided) or (**a**) two-way ANOVA for multiple comparisons.

## Discussion

Succinyl-CoA ligases (SUCLGs) and succinate dehydrogenases (SDH, electron transport chain (ETC) complex II) are functionally connected proteins involved in cellular mitochondrial energy production. SUCLGs regulate the generation of succinate as part of the TCA cycle and SDH oxidizes succinate into fumarate to accomplish electron transfer to the terminal acceptor ubiquinone in the ETC[31,32]. Mutations in SDH have been identified in various solid cancer types including familial paraganglioma/pheochromocytoma where inactivation of SDH results in epigenetic and metabolic alterations resulting in a build-up of TCA-cycle intermediates[33]. In AML, mutations in SUCLG or SDH are uncommon, although SDHB mutations were described in a case of chronic lymphocytic leukemia[34]. In our current work we identify that succinyl ligases are upregulated and that the ETC complex II is highly active in OXPHOS-driven AMLs that are frequently *FLT3*-ITD+, which can be therapeutically exploited.

Besides an upregulation of Succinyl Ligases, we find other factors that might further support the OXPHOS state of *FLT3*-ITD+ AMLs. This includes CD36 (Supplementary Fig. 1a), which has been reported to be upregulated in chemotherapy-resistant cells, thereby driving fatty-acid oxidation[35,36]. By performing proteome studies in functionally defined leukemia-initiating cell populations it was shown that *FLT3*-ITD+ LSCs are enriched for OXPHOS signatures[37] and the authors also identified both SUCLG1 and SUCLG2 to be higher expressed in the LSC subcompartment compared to blast population and also compared to the healthy HSPC population. Baccelli et al. have shown that the metabolic landscape of AML is heterogeneous[16], in line with our findings (current work and[22]), whereby *FLT3*-ITD+ AMLs are classified as OXPHOS^high. Sriskanthadevan et al. showed that in particular ETC complex II activity was enhanced in a panel of 11 primary AML samples compared to healthy counterparts[38]. It was also shown that treatment with venetoclax together with azacitidine disrupts ETC complex II-driven energy metabolism and eradicates persisting ROS^Low LSCs[39]. The mitochondrial peptidase neurolysin, NLN, has been shown to be an essential protein to control oxidative phosphorylation in a subset of AML patients and was upregulated in *FLT3*-ITD+ AMLs[40]. Also, in our proteome dataset, we find NLN to be higher expressed in *FLT3*-ITD+ AMLs (Supplementary Data 2). It was proposed that NLN is associated with an increased formation of respiratory chain supercomplexes[40], and although this requires further studies it is interesting to hypothesize that *FLT3*-ITD might impact on mitochondrial activity via similar mechanisms as well. Inhibition of the mitochondrial protease CLPP selectively kills AML cells by

inhibition of oxidative phosphorylation and mitochondrial metabolism[41] and we find CLPP to be higher expressed in *FLT3*-ITD+ AMLs (Supplementary Data 2). Together, these data indicate that the signaling networks downstream of *FLT3*-ITDs rewire metabolism at multiple levels that all facilitate a high OXPHOS state. We propose that factors such as STAT5 might play an important role, but which transcription factors downstream of *FLT3*-ITDs participate in these signaling networks is under current investigation, also in a larger cohort of AML patients at the subclonal level by performing transcriptome and chromatin accessibility studies, in line with our previous work[7]. However, we do not propose that an OXPHOS-driven metabolism is exclusive for *FLT3*-ITD+ AMLs. For instance, MLL-AF9+ THP1 AML cells demonstrated OXPHOS activities as high as *FLT3*-ITD+ cells[22], and also within the cohort of *FLT3*-wt primary AML patients there were a few that displayed high OXPHOS. These data again highlight the heterogeneous landscape of AML and challenges ahead lie in the detailed characterization of how specific combinations of (epi)genetic alterations impact on the metabolome.

A potential pitfall that is frequently encountered with single-agent treatment strategies is drug resistance. By inhibiting specific pathways, tumor cells can swiftly rewire signaling, and also metabolic dependencies can easily be rerouted. Here, we uncover that AML cells can change to lactate-fueled respiration, most notably when ETC complex II activity is inhibited. This provides an attractive possibility for combination treatment strategies, and we indeed show that inhibition of the lactate importer MCT1 greatly enhances sensitivity to ETC complex II inhibition, both in vitro as well as in vivo in xenograft mouse models. These observations are somewhat similar to what was shown in pheochromocytoma and paraganglioma tumors where SDH loss induced a vulnerability to LDH and pyruvate carboxylase inhibition[42]. In AML, it was also suggested that inhibition of lactate metabolism might be an attractive strategy[43]. Also in line, Cardaci et al. showed that pyruvate carboxylation is an adaptive mechanism for the survival of SDH-deficient kidney cells and their tracing studies showed that pyruvate consumption is higher in SDH-deficient cells[28]. Similar to our observation, succinate was built up in the absence of SDH and the authors show that the increased consumption of extracellular pyruvate diverts glucose-derived carbons into aspartate biosynthesis, thereby sustaining cell growth[28]. In a parallel study, Lussey-Lepoutre et al. also show that loss of SDH activity increases the dependency of cells on pyruvate carboxylation to generate aspartate and drive cellular anabolism[29].

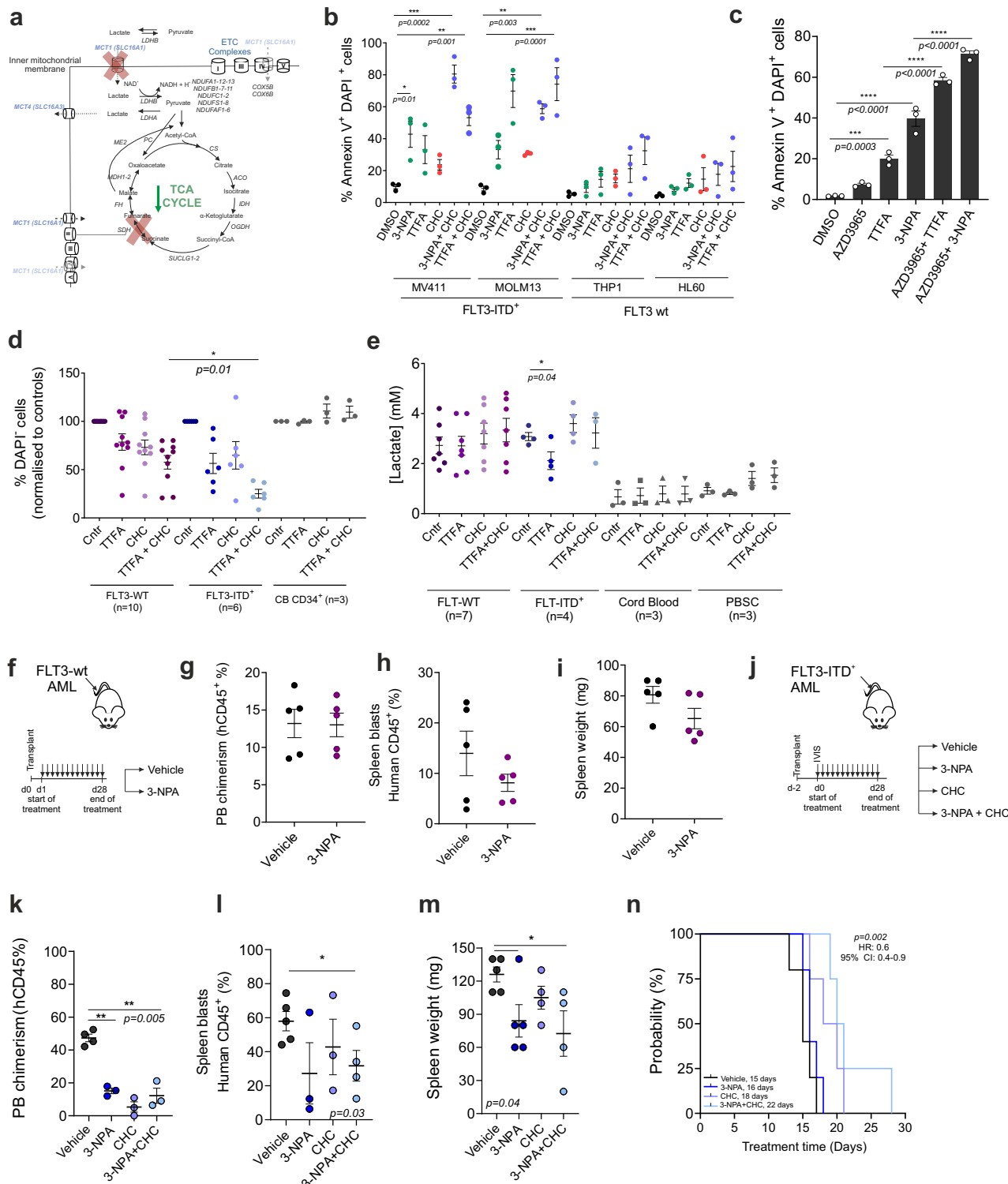

Overall, our results are in line with these previous observations in different cell types, whereby pyruvate and lactate seem to be able to play an important role to support the TCA cycle when ETC complex II activity is deficient or low. While we have not extensively studied whether extracellular pyruvate can also rescue the ETC complex II inhibitor-induced phenotypes as we have seen upon lactate supplementation, our tracing studies clearly indicate that lactate is converted into pyruvate and consecutive further downstream TCA intermediates. Similar to functional observations above, elegant computer-based simulations from

Zielinski et al. also predicted that ETC complex II deficient cells required maximal NADH and increased lactate oxidation to cross the mitochondrial inner membrane to support TCA-cycle activity and pyruvate was increasingly produced from lactate[30]. The mechanism behind this was indeed oxaloacetate production from increased pyruvate carboxylation and from malate generated from aspartate. An additional mechanism that was proposed in the same study was increased reductive carboxylation adaptive metabolism in SDH deficiency[30]. Our results from tracing studies suggest that lactate oxidation drives active cytosolic reductive

**Fig. 5 Combined inhibition of ETC complex II and lactate transporters impairs leukemogenesis in ITD$^+$ AMLs in vitro and in vivo. a** Schematic representation of the truncated TCA cycle and OXPHOS in *FLT3*-ITD$^+$ AMLs after ETC complex II and MCT1 transporters inhibition. **b** (%) Annexin V$^+$DAPI$^+$ AML cells upon ETC complex II inhibitors (TTFA 200 μM, 3-NPA 2 mM) with/without 800 μM CHC for 24 h (mean +/– SEM of three biological replicates (each dot)). **c** (%) Annexin V$^+$DAPI$^+$ MOLM13 cells upon ETC complex II inhibitors (TTFA 100 μM and 3-NPA 2 mM) with/without 50 nM AZD3965 for 24 h (mean +/– SEM of three biological replicates). **d** % DAPI$^-$ CD34$^+$ AML patient cells ($n = 6$ *FLT3*-ITD$^+$, $n = 10$ *FLT3*-wt) and in CD34$^+$ cord blood (CB) cells ($n = 3$) treated with TTFA 100 μM with/without 800 μM CHC for 48 h. Data normalized to controls. Each dot = independent patient, measured in technical triplicates (mean +/– SEM). **e** Extracellular lactate concentration [mM] in culture medium of cells screened in (**d**); CD34$^+$ AML patient cells ($n = 4$ FLT3-ITD$^+$, $n = 7$ FLT3-wt) and in healthy CD34$^+$ cord blood (CB) cells ($n = 3$) and peripheral blood mobilized stem cells (PBSC) treated with TTFA 100 μM with/without 800 μM CHC (48 h) (mean +/– SEM). (**f**) *FLT3*-wt HL60-transplanted mice were treated daily intraperitoneally with 3-NPA (10 mg/kg), ($n = 5$), or vehicle ($n = 5$). **g** (%) human CD45$^+$ cells in the peripheral blood (PB) and (**h**) in the spleen and (**i**) spleen weights at the time of sacrifice after 28 days of treatment (mean +/– SEM) ($n = 5$). **j** *FLT3*-ITD$^+$ MV411-transplanted mice were treated intraperitoneally daily (for a period up to 28 days when the last mouse succumbed to leukemia) with vehicle, 3-NPA (40 mg/kg), CHC (100 mg/kg) or with both ($n = 5$ in each group). **k** (%) human CD45$^+$ cells in the peripheral blood (PB) on day 14th of the treatment and (**l**) in spleen and (**m**) spleen weights at the time of sacrifice (mean +/– SEM) ($n = 5$). **n** Kaplan–Meier curve showing survival of mice transplanted with MV411 cells treated with the indicated therapy. The average survival in days are indicated at the bottom of the graph. *P* value (by log-rank test) indicate statistical significance for comparisons of 3-NPA + CHC treated mice with control mice (vehicle). **a**–**e**, **g**–**i** Student's *t* test (two-sided) or (**k-m**) one-way ANOVA Kruskal–Wallis test for multiple comparisons.

carboxylation activity in malate-aspartate-fumarate and oxaloacetate upon ETC complex II inhibition. How these lactate-driven intermittent pathways are beneficial apart from fueling mitochondrial respiration in AML cells remains unknown and requires further studies. In conclusion, we identify how cells reroute their metabolism in response to ETC complex II inhibition and identify lactate-fueled respiration as a targetable vulnerability in AML.

## Methods
**AML and healthy primary cell isolations and in vitro cell culture conditions**. Human myeloid cell lines used in this study were maintained in RPMI 1640 medium supplemented with 10% fetal calf serum (FCS) (Sigma-Aldrich), 1% penicillin and 1% streptomycin (Life Technologies, Grand Island, USA) at 37 °C and 5% CO$_2$. Neonatal cord blood (CB) samples were obtained from healthy full-term pregnancies and mobilized peripheral blood from healthy donors were obtained at the obstetrics departments at the Martini Hospital and University Medical Center Groningen. Bone marrow or peripheral blood samples were obtained of patients diagnosed with de novo AML. All human samples, including healthy CB and AML samples, were obtained and studied after informed consent and protocol approval by the Ethical Committee in accordance with the Declaration of Helsinki. The study was approved by the UMCG Medical Ethical Committee. Mononuclear cells (MNCs) were isolated via LymphoprepTM (Alere technologies, Oslo, Norway) density gradient-based separation and either used fresh or cryopreserved in our biobank until further use. When cryopreserved MNC fractions of AML patients and PB from allogeneic donors (PBMSC) were used, samples were thawed and then firstly resuspended in newborn calf serum (NCS) supplemented with DNase I (20 Units/mL), 4 mM MgSO$_4$ and heparin (5 Units/mL) and incubated on 37 °C for 15 min (min). Next, CD34$^+$ cells were isolated using autoMACS hematopoietic progenitor magnetic associated cell sorting kit automatically from Miltenyi Biotech according to the manufacturer's instructions. Primary AML cells were plated in Gartner's medium consisting of αMEM (Thermo Scientific) supplemented with 12.5% heat-inactivated fetal calf serum (Sigma-Aldrich), 12.5% heat-inactivated horse serum (Invitrogen), 1% penicillin and streptomycin (Life Technologies, Grand Island, USA), 57.2 μM β-mercaptoethanol (Merck Sharp & Dohme BV) and 1 mM hydrocortisone (Sigma-Aldrich), with the addition of 20 ng/mL G-CSF (Amgen), N-Plate (clinical grade TPO) (Amgen) and IL-3 (Sandoz). For primary AML co-cultures, mouse bone marrow-derived MS5 stromal cells were initially plated on gelatin-coated culture flasks and expanded to form a confluent layer, to which primary AML cells were added (unless indicated liquid culture). Liquid cultures and co-cultures were grown at 37 °C and 5% CO$_2$. Healthy CB and PBMSC were cultured in rich Stem line-S0192 (Sigma-Aldrich) medium with 1% penicillin and streptomycin-containing cytokines c-Kit (100 ng/ml) (255-SC, Novus Biologicals), FLT3-Ligand (100 ng/ml), N-Plate (100 ng/ml), and IL-3 (100 ng/ml).

**In vivo model using HL60 cells**. All experimental procedures were approved by the Institutional Animal Care and Use Committee of the University of Groningen and are in line with the Guide for Care and Use of Laboratory animals. Eight-week-old female NSG mice (005557, Charles River) were sublethally irradiated with a dose of 1 Gy, one day prior to transplantations. Mice were intravenously injected with $2 \times 10^5$ HL60 luciferase-GFP transduced cells. One day following AML transplant, the mice were randomly assigned to two groups ($n = 5$ per group) and daily treatment was initiated with vehicle and 3-NPA (Sigma-Aldrich; 10 mg per Kg) intraperitoneally. The 3-NPA was prepared weekly in 3.5% EtOH:PBS

Leukemic burden was monitored by peripheral blood chimerism (evaluation of the percentage of human CD45$^+$ cells). All mice were sacrificed at an onset time-point (day 28 of treatment) to determine differences in leukemic burden between treatment groups. Analysis was performed on the spleen, liver, and bone marrow.

**In vivo model using MV4–11 cells**. For AML xenografic models, 8-week-old female NSG mice (005557, The Jackson Laboratory) were pre-conditioned with 20 mg per kg of busulfan (Sigma-Aldrich) one day prior to the transplant. On the next day, $5 \times 10^5$ luciferase-transduced MV4–11 cells were intravenously injected, as indicated in the schematic for the murine model (Fig. 5k). After confirmation of engraftment (two days after transplant), the mice were randomly assigned to four groups ($n = 5$ per group) and treated with vehicle, 3-NPA (Sigma-Aldrich; 40 mg per kg; intraperitoneally), CHC (Sigma-Aldrich; 100 mg per kg; intraperitoneally) or a combination of these. The 3-NPA was prepared weekly in 3.5% EtOH:PBS. CHC was formulated weekly in 2% DMSO, 26% NaOH (1 mM), and 72% PBS. The leukemic burden of MV4–11 cells was monitored by bioluminescence imaging at day 0 and day 10, and by peripheral blood chimerism (evaluation of the percentage of human CD45$^+$ cells). Briefly, mice were anesthetized and injected intraperitoneally with the firefly luciferase substrate D-luciferin (100 mg per kg, PerkinElmer) and were then imaged with the IVIS Lumina system (Caliper Life Sciences Inc, Hopkinton, MA). Mice were followed for overall survival analysis and sacrificed when mice showed severe signs of illness or when tumor burden exceeded 40% of human cells in the peripheral blood of mice. At the end of the experiment, animals were harvested and subjected to analysis of spleen and bone marrow. Only mice with viable material (bone marrow or spleen) were included in the analysis of bone marrow counts and chimerism. Viable MV4–11 cells (Human CD45$^+$) in the spleen and bone marrow were evaluated by analysis of human CD11b and CD14 (BD biosciences) inside the population of negative CD45.1 (BD biosciences) by flow cytometry. All animals were housed under specific pathogen-free conditions in individually ventilated cages during the whole experiment and were maintained according to the Guide for Care and Use of Laboratory Animals of the National Research Council, USA, and to the National Council of Animal Experiment Control recommendations. All experiments were approved by the Animal Ethics Committee of the University of Sao Paulo (#067/2018).

**Proteome, GO, and GSEA studies**. Proteome data were derived from our previously published study[7]. There, the full label-free quantitative proteome was analyzed for 42 AML patient samples, and details regarding detection and quantification are outlined there. For one of the samples, the *FLT3* status was unknown, so in the current work 41 AML patient samples were included. For the majority of these, the CD34$^+$ fraction was analyzed, except for *NPMcyt* samples for which the blast population was analyzed ($n = 6$) (Supplementary Data 1). GO-analysis was performed using the online GeneOntology[1,44]. GSEA v4.1.0 on pre-ranked gene lists was performed with respect to MSigDB genesets C2 and C5 GO biological processes (version 7.3).

**Oxygen consumption and extracellular acidification rate measurements**. Oxygen consumption rate (OCR) and Extra Cellular Acidification Rate (ECAR) were measured using Seahorse XF96 analyzer (Seahorse Bioscience, Agilent, USA) at 37 °C. For AML cell lines and sorted CD34$^+$ from primary PBSCs and AML patients 100.000 and 200.000 viable cells (DAPI$^-$), respectively, were seeded per well in poly-L-lysine (Sigma-Aldrich) coated Seahorse XF96 plates in 180 μL XF Assay Medium (Modified DMEM, Seahorse Bioscience). For OCR measurements, XF Assay Medium was supplemented with 10 mM Glucose and 2.5 μM oligomycin A (Port A), 2.5 μM FCCP (carbonyl cyanide-4-(trifluoromethoxy) phenylhydrazone) (Port B) and 2 μM antimycin A together with 2 μM rotenone (Port C)

were sequentially injected in 20 μL volume to measure basal and maximal OCR levels (all reagents from Sigma-Aldrich). For ECAR measurements, a glucose-free XF Assay medium was added to the cells and 10 mM glucose (Port A), 2.5 μM oligomycin A (Port B), and 100 mM 2-deoxy-D-glucose (Port C) (all reagents from Sigma-Aldrich). For measuring the metabolic activity after TTFA, we initially treated the cells with 100 μM TTFA overnight and counted the remaining viable cells, and loaded equally with controls. For lactate-associated real-time OCR measurements, XF Assay Medium was supplemented with Glucose (10 mM) in which each injection mix is prepared. Final concentrations of 600 μM TTFA (Port A) with and without 600 μM CHC or DMSO, 2 mM sodium L-lactate (Sigma-Aldrich) (Port B, C, D) were sequentially injected in 10–20 μl volume to measure basal and maximal OCR levels. All XF96 protocols consisted of four times mix (2 min) and measurement (2 min) cycles, allowing for determination of OCR at basal and also in between injections. Both basal and maximal OCR levels were calculated by assessing the metabolic response of the cells in accordance with the manufacturer's suggestions. The OCR measurements were normalized to the viable number of cells used for the assay.

**Viability assays.** Primary healthy, AML samples and AML cell lines were cultured (as described in the in vitro cell culture conditions) with indicated medium and with and without inhibitors for 24 h or up to 2 days. Viability was assessed by DAPI staining and counted using a MACSQuant flow cytometer (Miltenyi Biotech) in a 96-wells format and analyzed by FlowJo v10.0.6 software (TreeStar, Ashland, OR). AML cell lines were used at 20.000 cells per well and CB and PBMSC cells were plated between 10.000–20.000 per well in 200 μL medium in 96-wells format.

**Lentiviral transductions.** HL60, TF1, and CB CD34+ cells used for transducing with empty vector and FLT3-ITD+ overexpression constructs using a lentiviral system. Lentiviruses containing a pRRL.SFFV.EGFP (gift from Christopher Baum) as a scrambled empty vector, MLL-AF9[45], or FLT3-ITD (pRRL.CMVp-FLT3.ITD 598-IRES-GFP, a gift from Renata Stripecke, Hannover, Germany) were generated using the Fugene transfection system (Promega, Madison, USA), together with packaging construct psPAX2 (Addgene, #12260) and glycoprotein envelope plasmid pMD2.G (Addgene, #12259). 293T cells cultured in RPMI (-FCS + 1% pen/strep) were transiently transfected in T75 flasks. Viral supernatant was harvested and filtered through a 0.45-μm filter after two days of the transfection and either used to transduce recipient cells (with 8 μg/mL polybrene to increase the infection efficiency) or stored at −80 °C for future usage. For transduction efficiency, GFP expression was checked using flow cytometry.

**ShRNA design.** Short hairpin RNA sequences used in this study are: Shluc(firefly): GGATTACCAGGGATTTCAGTC; shSUCLG1(#1): GGTACGAGTCAAGCACAAA CT; shSUCLG2 (#2): GCATCATGCCTGGCCATATTC; ShSUCLG2 (#1): GTAAAG GAAGCTCAAGTATAT; ShSUCLG2 (#2): GAATATAACTGGTATCCAAGA. The SUCLG1 hairpins were cloned in pLKO-GFP and the SUCLG2 hairpins cloned in pLKO-mBlueberry2. Lentiviral transductions are followed as described above. For quantitative RT-PCR, one microgram of total RNA (TRIzol LS reagent, Thermo Fisher) was reverse transcribed using the iScript cDNA synthesis kit (Bio-Rad) and amplified using SsoAdvanced SYBR Green Supermix (Bio-Rad) on a CFX384 Touch Real-Time PCR Detection System (Bio-Rad). Reactions were performed in duplicate, and in the case of coefficient of variation superior to 2%, a new reaction was performed. For SUCGL1 and SUCGL2 quantification, the RQ-PCR was performed with specific primers for each gene: SUCLG1 (succinate-CoA ligase GDP/ADP-forming subunit alpha, forward: AACTGCCCTGGAGTCATCAATC; reverse: CTGGACACA ATGCCAATCCTTC), SUCLG2 (succinate-CoA ligase GDP-forming subunit beta, forward: GGTGGTGTAAAGGAAGCTCAAG; reverse: GCAATGATGGCACAGTT GAC), the ACTB (Beta-actin, forward: AGGCCAACCGCAAGAAG; reverse: ACAG CCTGGATAGCAACGTACA) and RPL30 (Ribosomal protein L30, forward: ACTGC CCAGCTTTGAGGAAAT; reverse: TGCCACTGTAGTGATGGACAC), were used as housekeeping genes. The gene expression of the target genes was calculated relative to a reference cDNA (wild-type cell line) and set to 1. In all experiments, the same reference cDNA was used as an internal control to ensure that the results would be fully comparable among experiments. The gene expression values of the genes of interest were calculated as relative quantification using the ΔCt method and expressing the results as 2-ΔΔCt, in which ΔΔCt = ΔCt$_{sample}$ − ΔCt$_{cell line}$.

**Reagents.** TTFA (2-thenoyltrifluoraceton), rotenone, 3-nitropropionic acid, and MCT1 inhibitor (α-cyano-4-hydroxycinnamic acid) were purchased from Sigma-Aldrich. The second MCT1 inhibitor AZD3965 was purchased from MedChem-Express, NL. Stock solutions were prepared in following concentrations; 200 mM TTFA (in DMSO), 80 mM rotenone (in DMSO), 830 mM 3-Nitropropionic acid (%70 EtOH-PBS mix), α-cyano-4-hydroxycinnamic acid 200 mM (in 1 M NaOH-PBS mix), AZD3965 100 mM (in DMSO). 2, 2-dichloroacetophenone (DAP) was purchased from Sigma (St. Louis, MO) and stock solutions of the drug were prepared at 7 mM in dimethyl sulfoxide (DMSO). Sodium L-lactate and sodium pyruvate were purchased from Sigma-Aldrich (Cat# L7022 and Cat# S8636, respectively). Quizartinib (AC-220) was obtained from MedChemExpress (Cat# HY-13001).

**Labeled [13C3] lactate tracing with GC-MS.** A total of $15 \times 10^6$ MOLM13 cells were grown with or without 200 μM TTFA and with or without 8 mM and 20 mM uniformly labeled [U-13C3]-lactate (Cambridge Isotope Laboratories, Buchem B.V., NL) for 6 h. At the end of the experiment, cells were counted, media and cell pellets were separated. Cells were washed twice with ice-cold PBS before snap-freezing with liquid nitrogen and were stored at −80 °C until sample processing. We extracted polar intracellular metabolites from pellets (15 million cells/sample) using ice-cold MeOH:H$_2$O:CHCl$_3$ (1:1:2) extraction method in 16 mm diameter glass tubes. In total, 5 ng of norvaline was added to each sample as a recovery standard for relative quantification of metabolites. Tubes were shaken for 30 min at 4 °C and then were centrifuged for 15 min at 1600×g at 4 °C. The upper polar phase containing the polar metabolites was transferred to another tube and was dried under airflow. In total, 50 μl of DCM (dichloromethane) were added to each tube and dried with nitrogen gas to guarantee total dehydration of the sample prior to derivatization. The interphase containing proteins was also transferred to the another tube. The interphase was washed with cold MeOH, then dried under airflow and next disaggregated using 30% KOH solution at 100 °C for 15 min. Protein content was measured using BCA assay kit (Cat# 23227).

Derivatization of samples containing polar metabolites was run as follows: samples were firstly derivatized using 2% methoxamine hydrochloride in pyridine for 90 min with heating at 37 °C and shaking. As a follow-up, samples were incubated for 1 h at 55 °C with MBTSTFA + 1% TBDMCS and were then transferred to Gas Chromatography-Mass Spectrometry (GC-MS) vials. GC-MS analysis was run under electron impact ionization using a 7890A GC equipped with a HP5 capillary column coupled to an Agilent 5975C MS (Agilent, Santa Clara, CA, USA). For all GC-MS measurements, 2 μL of the sample was injected at 270 °C using pulsed split mode. GC oven temperature was held at 100 °C for 3 min after injection and increased to 300 °C through four consecutive ramp temperatures: first to 165 °C at 10 °C/min, next to 225 °C at 2.5 °C/min, next to 265 °C at 25 °C/min and finally to 300 °C at 7.5 °C/min. All spectra were recorded using full-scan MS configuration, from m/z 100 to 650. The Source File (Supplementary Fig. 5e) shows the retention time (RT) and m/z values that were monitored in our analysis. All integrations were performed manually, considering all the peak areas for each metabolite. Relative concentration of metabolites in cell pellets (metabolite ion counts) were normalized to norvaline ion counts as recovery standard and to the protein content. Midcor software was used to correct natural enrichment and ion interferences in the measured isotopologue distributions and to quantify 13C labeling incorporation and real isotopologue distributions[46].

**Oroboros assay.** Mitochondrial real-time respiration in AML cells was performed at 37 °C under continuous magnetic stirring with high-resolution respirometry with the O2k system (OROBOROS Instruments). Cell pellets were resuspended in 1 ml mitochondrial assay solution (mannitol and sucrose buffer: MAS; sucrose 35 mM, mannitol 110 mM, KH2PO4 2.5 mM, MgCl$_2$ 2.5 mM, HEPES 1.0 mM, EGTA 0.5 mM, fatty-acid free BSA 0.10%, pH 7.4) prior to respirometry. After permeabilization of AML cell lines ($2.5 \times 10^6$ cells) or AML primary CD34+-sorted cells ($3 \times 10^6$ cells) with 2.5 μg/ml digitonin in the chambers, we started sequentially injecting the following metabolites (with the following final concentrations) according to the selected experimental combination; 2 mM malate + 8 mM lactate, 1 mM ADP, 175 μM TTFA, MCT1 inhibitors (400 μM α-cyano-4-hydroxycinnamic acid or 150 nM AZD3965), 1.5 μM FCCP and finished the experiments with adding 0.05% sodium dithionite for calibrating instrumental O2 background. Basal respiration levels are corrected while comparing intermediate and final respiration values. Injections were followed every 3–5 min after respiration levels started to be stable. Assay analysis was performed using Datlab software (OROBOROS Instruments).

**ETC complex I and complex II activity measurements.** A total of $10 \times 10^6$ and $10–20 \times 10^6$ cells for AML cell lines and primary AML/healthy PBMSC CD34 + cells, respectively, with or without 100 μM TTFA for 24 h were collected, prepared and ETC complex II enzyme activity (Abcam, ab109908) and ETC complex I activity (Abcam, ab109721) were quantified according to the manufacturer's protocol. Samples were normalized according to the protein amount measured by BSA assay. Multiskan Sky microplate reader (Thermo Fisher Scientific) was used to detect colorimetric enzyme activity.

**Western blotting.** Cell pellets from AML cell lines, with $1 \times 10^6$ cells lines were lysed in RIPA buffer, and lysates were boiled at 100 °C for 5 min. The protein concentration of the samples was measured using the Pierce BCA protein kit (Cat#23227) and equal protein amounts (20 μg) were loaded into 4–15% mini Protean TGX precast gels for SDS-PAGE (Bio-rad Cat#4561085) and transferred to LF-PVDF membranes (Trans-blot Turbo RTA Transfer kit Cat# 1704274). Membranes were blocked in Odyssey blocking buffer (Cat # 927-40000) for 1 h and then incubated overnight with primary antibody mix Oxphos cocktail 1:3000 (Thermo Fisher Scientific Cat# 45-8199) and 1:3000 Actin (rabbit, Cell Signaling Technology, 4970). Membranes were washed three times with TBS-T and then incubated with the following secondary antibodies (1:3000); Alexa Fluor 680 goat anti-rabbit (Invitrogen, A21109), IRDye 800CW donkey anti-mouse (Li-Cor

926-32212). Scanning of the membranes was performed with an Odyssey Clx scanner (Li-Cor Biosciences).

**Spectrophotometric metabolite concentration assays**. The concentrations of glucose, lactate, glutamate, and glutamine in culture medium were determined by monitoring NAD(P)H increase occurring during specific enzymatic assays at 340 nm wavelength in a 96-wells plate. For real-time metabolite consumption/ secretion monitoring, cell culture supernatant was taken at 0 h, and after 10-24-36 h incubation with or without 100 μM TTFA, was collected. Lactate concentration was determined by the lactate dehydrogenase (LDH) enzymatic reaction in the cell culture medium. Extracellular lactate was converted by L-lactic dehydrogenase (LDH, Sigma-Aldrich) reaction in freshly prepared 25 mM NAD+ and 87.7 U/mL LDH in 0.4 M hydrazine (Sigma-Aldrich)/0.5 M glycine assay buffer (pH 9). The samples (20 μl; diluted according to standard curve) and sodium L-lactate (Sigma-Aldrich) standards were pipetted into 130 μl reagent mix in a 96-wells plate and the reaction was carried out for 30 min at 37 °C. Glucose consumption was detected using an enzymatic reagent mix consisting of 75 μl 100 mM PIPES buffer, 2.5 μl 40 mM NADP, 2 μl 10 mM ATP, 1 μl 500 mM MgSO₄, 0.15 μl hexokinase, 0.15 μl glucose-6-phosphate-dehydrogenase, and 44.3 μl H₂0 per well was prepared and 125 μl added to both medium and glucose standard wells. The plate was incubated for 30 min at 37 °C. The concentration of extracellular glutamine was determined by its conversion first to glutamate through the glutaminase (GLS) reaction followed by the quantification of glutamate concentration. GLS reaction was carried out for 30 min at 37 °C with shaking by adding 20 μl sample of culture media to 180 μl reaction mix consisted of 10 U/ml GLS (Sigma-Aldrich, Cat#G8880) in 0.5 M acetate buffer, pH 5. Determination of glutamate concentration was performed through the glutamate dehydrogenase (GLDH, Sigma-Aldrich) reaction at 37 °C for 30 min by adding 20 μl sample of culture media to 180 μl reaction mix (25 mM ADP, 40 mM NAD + and 100 U/mL of GLDH in 0.5 M glycine/0.5 M hydrazine buffer, pH 9). Glutamine concentration was calculated by deducting basal glutamate concentration from the final converted glutamate measurement. Multiskan Sky microplate reader (Thermo Fisher Scientific) spectrophotometer was used to detect the absorbance. Values of consumption and release of extracellular metabolites for each sample were normalized by cell number and incubation time considering the exponential growth curve.

**Flow cytometry and sorting procedures**. Prior to antibody staining, cells were blocked with anti-human FcR Block (Mylteni Biotech) and murine cells were blocked with anti-Fc (BD Biosciences). Samples were washed twice in PBS before each flow cytometric measurement. Fluorescence was measured on the MACS-Quant Analyzer 10 (Miltenyi Biotech). Apoptosis was quantified with Annexin V staining (FITC) according to the manufacturer's protocol (0.5 μL/100 μL) (Mylteni Biotech). An equal number of cells are stained with Annexin V in calcium supplied sterile water buffer at a concentration of $0.25 \times 10^6$ cells/ml for 20 min at 4 °C in the dark followed by a 1× wash and DAPI staining for 10 min. Cell sorting for transduced AML cell lines GFP+ isolation was performed on a MoFlo-Astrios (Beckman Coulter). Flow cytometric analyses, cell counting, and viability measurements were determined on MACSQuant (Miltenyi Biotech) flow cytometer. All flow cytometry data were analyzed using FlowJo v10.0.6 software (TreeStar, Ashland, OR). For in vivo studies, viable HL60 and MV4–11 cells (Human CD45+) in the spleen, peripheral blood, liver, and bone marrow were evaluated regarding the expression of human CD45+ and CD45.1− (CD45-PerCP Biolegend Cat# 3040256 (2 μL/100 μL) and CD45.1 PE Murine Cat#553776 (2 μL/100 μL), BD biosciences).

**mtDNA measurement**. Total DNA was isolated from total bone marrow cells ($2 \times 10^6$ cells) with the PureGene® kit (Qiagen, USA) according to the manufacturer's protocol. DNA concentration was measured with a Nanodrop™ spectrophotometer (Thermo Scientific) and 10 ng of DNA was used for each reaction. mtDNA copy number was performed by quantitative RT-PCR. Briefly, MT-ND4 and MT-CYB genes were used to represent the mtDNA, and pyruvate kinase (PK) and Hemoglobin Subunit Beta (HBB) genes were used to represent the nuclear DNA. Relative mtDNA copy numbers were assessed after MT-ND4/CYB normalization by the single-copy nuclear genes PK/HBB. All reactions were performed using SsoAdvanced SYBR Green Supermix (Bio-Rad) on a CFX384 Touch Real-Time PCR Detection System (Bio-Rad). The corresponding real-time PCR efficiencies for each mitochondrial and nuclear gene amplification were calculated according to the equation: $E = 10^{(-1/slope)} - 1$. The relative mtDNA copy number was calculated relative to a reference DNA sample (healthy donor) and set to 1. In all experiments, the same reference DNA (a hematological healthy donor) was used as an internal control to ensure that the results would be fully comparable among experiments. The relative mtDNA copy number was calculated as relative quantification using the ΔCt method and expressing the results as $2^{-\Delta\Delta Ct}$, in which $\Delta\Delta Ct = \Delta Ct_{sample} - \Delta Ct_{reference\ DNA}$.

**Quantification and statistical analyses**. The PREdiction of Clinical Outcomes from Genomic Profiles (PRECOG)[23] was used to evaluate the association between genes involved with metabolic pathways with overall survival in AML patients. PRECOG comprehends 08 different AML transcriptomic studies that included

patients diagnosed with de novo AML, with age superior to 18 years old. All the patients included in the studies were treated with curative intent, according to the Dutch-Belgian Hematology-Oncology Cooperative Group and the Swiss Group for Clinical Cancer Research. Statistical analyses were performed considering the expression levels of the studied genes as continuous variables.

For the LFQ quantitative proteome study, differentially expressed proteins between FLT3-ITD and FLT3-wt AMLs were identified using the online Galaxy Limma-Vroom tool (https://usegalaxy.org/).

Statistical significance of the difference between multiple experimental groups was analyzed by one-way or two-way ANOVA (Kruskal–Wallis test with post hoc Dunn analysis) using GraphPad PRISM 8 software or two-tailed paired or unpaired Student's t test using excel and GraphPad PRISM 8 software. Linear regression analysis was performed using GraphPad PRISM 8 software. P values are indicated in Fig. legends.

For comparisons between the different groups of treated mice, Kruskal–Wallis test (with Dunn's post hoc) was used to compare continuous variables. The cutoff used to define engraftment in the peripheral blood or bone marrow of the leukemic mice was the presence of human cells (CD45+) ≥ 0.1. Overall survival was plotted using Kaplan–Meier plots, using Cox proportional hazard regression to compare the differences between the curves, providing the Hazard ratio (HR) and the 95% confidence interval (CI). All P values were two-sided with a significance level of 0.05. All calculations were performed using the statistical package for the social sciences (SPSS) 19.0 and GraphPad Prism 8 software.

**Reporting summary**. Further information on research design is available in the Nature Research Reporting Summary linked to this article.

## Data availability
All data are also available from the corresponding author on request. LFQ proteome data are provided as Supplementary Data 2 and is deposited under PRIDE PXD030463. Furthermore, the following publicly available datasets were used: GSE13159 (Mile); GSE1427; GSE10358; GSE12417; and TCGA Cancer Genome Atlas Research (https://www.cancer.gov/about-nci/organization/ccg/research/structural-genomics/tcga). The remaining data are available in the Article and Supplementary Information. Source data are provided in this paper.

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

## Acknowledgements

This work was supported by a grant from the EU (ITN-EJD HaemMetabolome (H2020-MSCA-ITN-2015 675790) awarded to J.J. Schuringa and to M. Cascante. AE and AC gratefully acknowledge receipt of a Marie Curie Fellowship and are participants in the same Initial Training Network. D.A.P.M. received a fellowship from FAPESP (Grant #2017/23117-1). M. Cascante and S. Marin also acknowledge support from MICINN (SAF2017-89673-R, AEI/FEDER, UE), from AGAUR (2017SGR1033), and from ISCIII (CB17/04/00023). M.C. is also supported by ICREA Academia Program.

## Author contributions

Conceptualization: A.E. and J.J.S; investigation: A.E., S.M., A.C., D.A.P.M., E.M.R., G.H., M.C., and J.J.S.; technical assistance and discussion: M.G., M.P., I.W., A.G., B.M.B., and A.T.J.W.; resources: G.H.; data curation: A.E., S.M., M.C., and J.J.S.; writing—original draft: A.E. and J.J.S; writing—review and editing: A.E., S.M., A.C., D.A.P.M., M.G., M.P., I.W., A.G., B.M.B, A.T.J.W., G.H., M.C., and J.J.S.; funding acquisition: M.C. and J.J.S.; overall supervision: M.C. and J.J.S.

## Competing interests

The authors declare no competing interests.
