## [Peer Review File · Nature Communications]

Inhibition of the succinyl dehydrogenase complex in Acute Myeloid Leukemia leads to a lactate-fuelled respiratory metabolic vulnerabilityREVIEWER COMMENTS

Reviewer #1 (Remarks to the Author):

In this study Erdem and colleagues study the role of OXPHOS in AML, uncovering that activating mutations of the RTK FLT3 drive an OXPHOS phenotype. Using quantitative proteomic analysis of multiple patient samples they then mechanistically link this to high expression of succinate-CoA ligases and activity of the mito ETC complex II. Further experiments demonstrate that inhibition of the mito ETC complex II is a vulnerability of this type of AML, but that mitochondrial respiration can be rescued by utilisation of lactate where this acts as a carbon source to anaerobically maintain the TCA cycle. Inhibition of the mitochondrial import of lactate by MCT1 inhibitors prevented this rescue in vitro and finally dual inhibition of MCT1 and ETC complex II was shown to kill leukaemia cells both in vitro and in vivo, where survival was increased in xenotransplant models.

This is an interesting and elegant multidisciplinary study, that combines complicated metabolomic, proteomic experiments +/- perturbations with relevant functional experiments in vitro and in vivo. A major strength is that the majority of experiments are performed in primary patient samples, and the subclonal experiments of the FLT3+ and negative clones from the same patient are particularly elegant in demonstrating the genotype specific effects. Despite this there are still areas that need further attention and clarification.

General

1. The continual reference to another submitted paper on PDK1 is somewhat distracting, as the data are not shown, thus we rely on the authors views without the substantiation of the data. In revision, this should be addressed, for example, I do not think that many of the references are necessary for this particular story.

Specific

1. The association between SUGLC1/2 levels and FLT3 mutation is compelling, but I note that some FLT3 WT samples have higher levels of proteins. Are there any further genotype associations of a high SUGLC1/2 level in FLT3 WT patients?
2. For Figure 1i, how much of the prognostic value lies with the single genes and how much is purely an association with FLT3 mutations status? I take it that this is univariate analysis? When multivariate analysis is performed taking into account age, WCC and FLT3 mutations status, how many of these positive HR remain?
3. In Figure 2d the cell lines seem to segregate more on MLL-rearrangement status than FLT3 mutational status. Do the authors have any further data to link MLL expression to SUGLC1/2 levels, ETC complex II activity and OXPHOS status? They allude to this in the discussion, but for example does retroviral overexpression of MLL-AF9 alter OCR, SUGLC1/2 levels or ETC complex II activity. Or to add to point 1 does the presence of an MLL fusion confer higher SUGLC1/2 levels in patient samples?
4. I do not understand the comment on page 9 regarding increased actin levels in CD25+ cells. Surely if equal protein amounts were loaded this should correct for minor changes in cell size between CD25+ and -ve fractions?

Reviewer #2 (Remarks to the Author):

The manuscript by Erdem et al. shows that FLT3-ITD+ AML specimens have high levels of oxidative phosphorylation that is driven in part by high activity of succinate-CoA ligases. Further, inhibition of ETC Complex II results in decreased oxidative phosphorylation which can be rescued by lactate. In FLT3-IDT AML lactate is taken up by the cells and used to synthesize TCA cycle intermediates and supports oxidative phosphorylation. Lactate transport inhibitors potentiate the effect of ETC complex II inhibitors in FLT3-ITD AML. This manuscript has many strengths including being novel, therapeutically relevant and complementary use of primary AML specimens and cell lines. Some additional areas could be addressed to further strengthen the manuscript.

Major Comments:

1. What are the metabolic consequences of knocking down succinate-CoA ligases in FLT3-ITD AML and FLT3 WT AML?
2. In figure 5b and 5c TTFa treatment was done for 24 hours which the authors show is a sufficient amount of time to kill cells (shown in 3a). This makes it difficult to determine if the changes observed are due to dead/dying cells. These experiments should be repeated at an earlier timepoint (s) to determine if a decrease in OCR and ECAR occur prior to cell death. Looking at % dead cells simultaneously would help ensure metabolic changes are not due to cell death.
3. Does the addition of lactate with and without CHC change OCR in FLT3 WT cells? (Figure 6a and b)
4. For in vivo experiments (Figure 7g-n) what was the percent of leukemic cells in the bone marrow at the end of treatment?
5. What did the luciferase signals in the MV4;11 in vivo model look like over time?

Minor Comments:

1. Do the authors have any thoughts or explanation for what looks like two populations in FLT3 WT SLUG2 expression?
2. The data presented in Figure 1 shows a clear correlation between protein levels and metabolism. However, the statement "These data indicate that the metabolic state of AML cells is, at least in part, instructed by their protein expression profile." is claiming a functional relationship which not completely supported by the data in Figure 1. It could be that metabolism drives changes in proteins levels.
3. In Figure 2j it is difficult to distinguish ETC complex IV and I.
4. In figure 5i having greater than 100% live cells due to the normalization process is counterintuitive. The authors may consider graphing this data in an alternative manner.
5. The diagrams in Figure 7 indicate that mice were treated for 28 days but the combination treatment group only lived for 22 days. It would be help for the authors provide an explanation for this discrepancy.

Reviewer #1 (Remarks to the Author):

In this study Erdem and colleagues study the role of OXPHOS in AML, uncovering that activating mutations of the RTK FLT3 drive an OXPHOS phenotype. Using quantitative proteomic analysis of multiple patient samples they then mechanistically link this to high expression of succinate-CoA ligases and activity of the mito ETC complex II. Further experiments demonstrate that inhibition of the mito ETC complex II is a vulnerability of this type of AML, but that mitochondrial respiration can be rescued by utilisation of lactate where this acts as a carbon source to anaerobically maintain the TCA cycle. Inhibition of the mitochondrial import of lactate by MCT1 inhibitors prevented this rescue in vitro and finally dual inhibition of MCT1 and ETC complex II was shown to kill leukaemia cells both in vitro and in vivo, where survival was increased in xenotransplant models. This is an interesting and elegant multidisciplinary study, that combines complicated metabolomic, proteomic experiments +/- perturbations with relevant functional experiments in vitro and in vivo. A major strength is that the majority of experiments are performed in primary patient samples, and the subclonal experiments of the FLT3+ and negative clones from the same patient are particularly elegant in demonstrating the genotype specific effects. Despite this there are still areas that need further attention and clarification.

General

1. The continual reference to another submitted paper on PDK1 is somewhat distracting, as the data are not shown, thus we rely on the authors views without the substantiation of the data. In revision, this should be addressed, for example, I do not think that many of the references are necessary for this particular story.

We thank the reviewer for her/his constructive comments and suggestions and appreciate the overall positive feedback. We fully understand the incomplete descriptions regarding the role of glycolytic regulator-PDK1 in different AML subtypes in this manuscript. In an independent manuscript that was submitted in parallel and now provisionally accepted in Nature Communications, we find that PDK1^{high} AMLs are typically FLT3-wt and largely glycolysis driven. PDK1 inhibition revealed selective targeting of FLT3-wt AMLs while OXPHOS-driven FLT3-ITD AMLs remained largely insensitive. In the current manuscript, we find OXPHOS inhibition has no major impact on targeting FLT-wt AMLs. So, although these two manuscripts reveal the heterogeneity within the metabolic landscape of AML in-depth independently, we made several cross-references between these two manuscripts at the time of submission. We now removed the links and cross-referencing in the current manuscript which solely focuses on FLT3-ITD OXPHOS metabolic signatures and adaptive behaviors.

Specific

1. The association between SUGLC1/2 levels and FLT3 mutation is compelling, but I note that some FLT3 WT samples have higher levels of proteins. Are there any further genotype associations of a high SUGLC1/2 level in FLT3 WT patients?

This is an excellent point, and based on the comments of the reviewer we have investigated this further and new data is added to Supplementary figure 1. We observed that the FLT3wt subgroup with high levels of SUGLC2 is indeed genetically distinct from the FLT3wt group with low levels of SUGLC2. In our proteome, we observed that all NPMcyt and inv16 AML subtypes resided within the

FLT3wt/SUCLG2high group (Supplementary Figure 1a). Pearson coefficients were calculated using the quantitative proteome comparing the FLT3-WT/SUCLG2high group with the FLT3-WT/SUCLG2low group, and the ranked list was used for GSEA which indicated significant enrichment for NPM1mut signatures, inv16 signatures and MLL-rearranged AML signatures (Supplementary Figure 1b). Next, we independently confirmed these data using the MILE AML transcriptome dataset (GSE13159) and observed that indeed the inv16 and MLL-rearranged AML subtypes express significantly higher levels of SUCLG2 compared to other AML subgroups.

2. For Figure 1i, how much of the prognostic value lies with the single genes and how much is purely an association with FLT3 mutations status? I take it that this is univariate analysis? When multivariate analysis is performed taking into account age, WCC and FLT3 mutations status, how many of these positive HR remain?

We thank the reviewer for the comment. In fact, all the p-values described are a product of a Cox Proportional Hazard Model considering Sex (categorical), Age (as a continuous variable), FLT3-ITD mutational status (mutant versus wild-type) and EuropeanLeukemia Net 2010 risk stratification (Favourable versus Intermediate I and II versus Adverse) (when available) as confounder variables. This information was properly included in the revised version of the manuscript (Results section, page #x, x paragraph, sentence in red). We performed the interaction analysis, confronting the expression of SUCLG1 and SUCLG2 with other prognostic effects, as suggested by you: age, WBC and FLT3-ITD mutational status. Using the TCGA cohort, which contain most of the confounder variables available, our results demonstrated that for the WBC (HR: 1.006; 95% CI: 1.001 – 1.011; P = 0.011; for the interaction) and age (HR: 1.024; 95% CI: 1.005 – 1.043; P = 0.012; for the interaction) the expression of SUCLG1 and SUCLG2 can interact in order to predict the outcome, but not with the FLT3-ITD mutational status (P = 0.488).

3. In Figure 2d the cell lines seem to segregate more on MLL-rearrangement status than FLT3 mutational status. Do the authors have any further data to link MLL expression to SUGLC1/2 levels, ETC complex II activity and OXPHOS status? They allude to this in the discussion, but for example does retroviral overexpression of MLL-AF9 alter OCR, SUGLC1/2 levels or ETC complex II activity. Or to add to point 1 does the presence of an MLL fusion confer higher SUGLC1/2 levels in patient samples?

As suggested by the reviewer, we transduced healthy cord blood (CB) CD34+ cells with both FLT3-ITD and MLL-AF9 over-expression constructs and evaluated their metabolic profiles. Similar to high OCR levels in FLT3-ITD and MLL-AF9 cell lines depicted in Figure 2d, we observed an increase in OCR upon transduction of FLT3-ITD and MLL-AF9 in healthy CB CD34+ cells (Supplementary Figure 2e). ECAR levels were slightly lower in FLT-ITD+ CB cells similar to what we observe in cell lines and primary patient samples. MLL-AF9+ CB cells showed an increase in ECAR, which we did not observe in the THP-1 cell line model. Moreover, we show that SUCLG2 and SDHA mRNA expression is significantly higher in FLT3-ITD+ transduced CB model but not in MLL-AF9-transduced CB model (Supplementary Fig, 2b).

We observe that Complex II and Complex I enzymatic activities are restricted to ITD+ (Figure 3f,h,i). We require around 20x10⁶ cells for measuring Complex II and I activities thus, technically it was challenging for us to screen CB cells after the transduction. However, when dichotomize AML patients regarding SUCLG2 protein levels (Sup fig.

1) we observe in GSEA analysis that *SUCLG2*^{high} *FLT3*-wt patients with are enriched for *MLLAF9* and *SUCLG2*^{low} for 8-21 translocations.

4. I do not understand the comment on page 9 regarding increased actin levels in CD25+ cells. Surely if equal protein amounts were loaded this should correct for minor changes in cell size between CD25+ and -ve fractions?

In line with comments of the reviewer, we were initially also a bit surprised that actin expression was different when we loaded either equal cell numbers or equal protein amount. But cells are indeed substantially different in cell size, which can also be observed from the FSC/SSC flow cytometry plots added below. Why despite correction for total protein content there were still slight differences in actin is not understood, but this is what the remark in the text referred to.

Figure for the reviewers. Differences in cell size between CD25- and CD25+ subclones. The CD25+ (in red) and CD25- (in blue) populations from the viable or blast gate were back-gated to FSC and SSC to show the cell size distributions.

Reviewer #2 (Remarks to the Author):

The manuscript by Erdem et al. shows that *FLT3*-ITD+ AML specimens have high levels of oxidative phosphorylation that is driven in part by high activity of succinate-CoA ligases. Further, inhibition of ETC Complex II results in decreased oxidative phosphorylation which can be rescued by lactate. In *FLT3*-ITD AML lactate is taken up by the cells and used to synthesize TCA cycle intermediates and supports oxidative phosphorylation. Lactate transport inhibitors potentiate the effect of ETC complex II inhibitors in *FLT3*-ITD AML. This manuscript has many strengths including being novel, therapeutically relevant and complementary use of primary AML specimens and cell lines. Some additional areas could be addressed to further strengthen the manuscript.

Major Comments:

1. What are the metabolic consequences of knocking down succinate-CoA ligases in FLT3-ITD AML and FLT3 WT AML?

We thank the reviewer for overall positive feedback and her/his suggestions and comments which allowed us to improve the current version of the manuscript. As suggested by the reviewer, we evaluated SUCLG1/2 activity in AML subtypes by using SUCLG1 and SUCLG2 directed shRNAs to downregulate their activity in both FLT3-wt (HL60 and K562) and FLT3-ITD AMLs (MOLM13 and MV411) (Supplementary Figure 2g). We assessed the consequences on the cell growth and the metabolic activity of the cells in both mitochondrial respiration and glycolysis states. Downregulation of both SUCLG1 and SUCLG2 significantly reduced the growth of FLT3-ITD⁺ MOLM13 and MV411 cells (Figure 3g, Supplementary Fig. 2h.), while proliferation of FLT3-wt HL60 and K562 cells was not reduced after the knockdown (Figure 3h, Supplementary Fig. 2i). Mitochondrial respiration activity of FLT3-ITD⁺ cells was significantly reduced (Figure 5c) which was also reflected in reduced glycolysis activity (ECAR measurement as lactate efflux by Seahorse) (Figure 5e), validating the effects of the TTFa inhibitor. Mitochondrial respiration activity and glycolysis activity remained comparable between shLuc and shSUCLG1/2 in FLT3-wt K562 cells (Supplementary Fig. 2k). Additionally, we evaluated the effect of ETC complex II inhibitor (TTFa)-induced apoptosis levels in AML cells after SUCLG1/2 knockdown and showed that sensitivity against TTFa was significantly reduced after SUCLG1/2 down-regulation (Figure 3i, 3j).

2. In figure 5b and 5c TTFa treatment was done for 24 hours which the authors show is a sufficient amount of time to kill cells (shown in 3a). This makes it difficult to determine if the changes observed are due to dead/dying cells. These experiments should be repeated at an earlier timepoint (s) to determine if a decrease in OCR and ECAR occur prior to cell death. Looking at % dead cells simultaneously would help ensure metabolic changes are not due to cell death.

We thank the reviewer for raising this point and giving us chance to discuss in more detail. In Fig.3a and 3c we treated AML cell lines and primary patient samples with 200 μ M for 24hrs and 48hrs, respectively. For metabolic activity measurements in Figure 5b and 5c, we used 100 μ M TTFa pre-treatment for 24 hrs and counted viable cell amounts to be able load equal number of cells to seahorse 96-wells plate prior to the measurement. As suggested by the reviewer, we treated AML cells with 100 μ M TTFa for 6 hours and assessed the impact of TTFa on mitochondrial respiration. Although mitochondrial respiration was already significantly reduced after 6 hours of 100 μ M TTFa treatment, apoptosis levels were not yet increased (Supplementary Fig.2j).

3. Does the addition of lactate with and without CHC change OCR in FLT3 WT cells? (Figure 6a and b).

Previously, we assessed the impact of lactate addition together with malate and ADP in FLT3-wt cell lines and FLT3-wt primary AML patient samples (Supplementary Figure 3e) in the presence of CHC and TTFa using the Oroboros assay. As suggested by the reviewer, we have now assessed the role of lactate in FLT3-wt AMLs in more detail. We repeated the experimental set-up used in Figure 6a and b in FLT3-wt HL60 cells using the Seahorse assay. Firstly, we observe that addition of TTFa did not reduce OCR in HL60 cells. The addition of lactate in the absence of CHC slightly increased respiration of HL60 cells and the addition of CHC and TTFa together slightly reduced OCR levels (Supplementary Fig 3a). Moreover, we performed Oroboros assays with

HL60 FLT3 WT cells similar to what was done for MOLM13 cells in Figure 6c. Here, we did not observe significant effects of lactate import inhibition on respiration (Supplementary Fig. 3c and 3d for representative real-time data.)

4. For in vivo experiments (Figure 7g-n) what was the percent of leukemic cells in the bone marrow at the end of treatment?

We assessed the leukemic burden in bone marrow by evaluating leukemic cell number per femur and by checking human mtDNA (mitochondrial DNA) content and added this information in the current version of our manuscript. Human mtDNA content from the peripheral blood was significantly positively correlated with the luciferase intensity measurement in both HL60 and MV411 transplanted mice (Supplementary Figure 5c). We have showed that bone marrow human mtDNA content was significantly reduced after co-treatment with 3-NPA and CHC (Supplementary Figure 5e) at the time of sacrifice indicating reduced tumour burden. Finally, the leukemic cell counts per femur were significantly reduced in the combination treatment group compared to vehicle group (Supplementary Figure 5h).

5. What did the luciferase signals in the MV4;11 in vivo model look like over time?

We have now included comparison of luciferase measurements at the start and at day 10 of the therapy in different treatment groups: vehicle, 3-NPA, CHC or combination of both (Supplementary Figure 5b and 5f). We observed that the combination treatment of 3-NPA together with CHC further reduced luciferase intensity compared to vehicle group.

Minor Comments:

1. Do the authors have any thoughts or explanation for what looks like two populations in FLT3 WT SLUG2 expression?

This is an excellent point that was also raised by reviewer 1. We observed that the FLT3wt subgroup with high levels of SUCLG2 is indeed genetically distinct from the FLT3wt group with low levels of SUCLG2. In our proteome, we observed that all NPMcyt and inv16 AML subtypes resided within the FLT3wt/SUCLG2high group (Supplementary Figure 1a). GSEA on pearson correlation coefficients ranking SUCLG2 expression with our AML quantitative proteome dataset further indicated significant enrichment for NPM1mut signatures, inv16 signatures and MLL-rearranged AML signatures (Supplementary Figure 1b). Next, we independently confirmed these data using the MILE AML transcriptome dataset (GSE13159) and observed that indeed the inv16 and MLL-rearranged AML subtypes express significantly higher levels of SUCLG2 compared to other AML subgroups.

2. The data presented in Figure 1 shows a clear correlation between protein levels and metabolism. However, the statement “These data indicate that the metabolic state of AML cells is, at least in part, instructed by their protein expression profile.” is claiming a functional relationship which not completely supported by the data in Figure 1. It could be that metabolism drives changes in proteins levels.

We agree with the reviewer that the proteome data cannot directly be causally related to metabolic states, and we have now removed this statement form the text.

3. In Figure 2j it is difficult to distinguish ETC complex IV and I.

We have repeated the Western blot in Figure 2j to be able to clearly display differences in ETC complex IV and I corresponding bands. The new representative image of biological replicates is now replaced in Figure 2j.

4. In figure 5i having greater than 100% live cells due to the normalization process is counterintuitive. The authors may consider graphing this data in an alternative manner.

We thank for this comment, Figure 5j has been corrected now and shows fold changes in viable cells after normalization to the controls.

5. The diagrams in Figure 7 indicate that mice were treated for 28 days but the combination treatment group only lived for 22 days. It would be help for the authors provide an explanation for this discrepancy.

We apologize for the confusion. The diagrams in fig 7k indicate the maximum treatment time, this was only reached in the combination treatment group in one mouse as shown in the Kaplan Meijer curve in Fig.7o. The average treatment periods are shown in the legend, and the average survival of mice in the combination treatment group was 22 days. This has now been clarified in the text.

REVIEWER COMMENTS

Reviewer #1 (Remarks to the Author):

The authors have answered all of my concerns and in doing so have improved the manuscript. I am happy that it is now of a quality for publication in Nature Communications

Reviewer #2 (Remarks to the Author):

The authors addressed all of my concerns.